# Spatial separation between replisome- and template-induced replication stress signaling

Néstor García-Rodríguez[1] , Magdalena Morawska[1,2,†], Ronald P Wong[1] , Yasukazu Daigaku[2,‡] & Helle D Ulrich[1,*]

## Abstract

Polymerase-blocking DNA lesions are thought to elicit a checkpoint response via accumulation of single-stranded DNA at stalled replication forks. However, as an alternative to persistent fork stalling, re-priming downstream of lesions can give rise to daughter-strand gaps behind replication forks. We show here that the processing of such structures by an exonuclease, Exo1, is required for timely checkpoint activation, which in turn prevents further gap erosion in S phase. This Rad9-dependent mechanism of damage signaling is distinct from the Mrc1-dependent, fork-associated response to replication stress induced by conditions such as nucleotide depletion or replisome-inherent problems, but reminiscent of replication-independent checkpoint activation by single-stranded DNA. Our results indicate that while replisome stalling triggers a checkpoint response directly at the stalled replication fork, the response to replication stress elicited by polymerase-blocking lesions mainly emanates from Exo1-processed, postreplicative daughter-strand gaps, thus offering a mechanistic explanation for the dichotomy between replisome- versus template-induced checkpoint signaling.

**Keywords** DNA damage bypass; DNA damage checkpoint; Exo1; postreplication repair; replication stress
**Subject Categories** DNA Replication, Repair & Recombination
**The EMBO Journal (2018) 37: e98369**

## Introduction

Genome maintenance relies on checkpoint pathways that perceive DNA damage or replication problems and initiate an appropriate response. In eukaryotic cells, they are mediated by kinase cascades activated by distinct types of abnormal DNA structures (Nyberg et al, 2002). In vertebrates, damage signaling by the ATM kinase is initiated by DNA double-strand breaks (DSBs), whereas the related ATR kinase reacts to a variety of lesions and is activated mainly by single-stranded DNA (ssDNA). During S phase, cells are particularly vulnerable to conditions that challenge the progression of the replisome. In this situation, ssDNA is thought to accumulate at stalled replication forks by an uncoupling between helicase and polymerase movement or between leading and lagging strand synthesis. In budding yeast, the checkpoint response elicited by these structures is initiated by Mec1, the homologue of vertebrate ATR, which is responsible for activating an effector kinase, Rad53. Via phosphorylation of a large set of substrates, Rad53 mediates most aspects of the checkpoint response, including a stabilization of stalled forks, suppression of late origin firing, control of nucleotide levels, regulation of damage-induced transcription, and arrest of the cell cycle (Pardo et al, 2017). Intriguingly, checkpoint signaling in response to replication stress can be divided into two branches that both initiate from Mec1 and converge on Rad53, but differ in the mediator protein responsible for signal transmission: the DNA replication checkpoint and the DNA damage checkpoint (Pardo et al, 2017). Upon inhibition of ribonucleotide reductase by hydroxyurea (HU), Mec1 phosphorylates the replisome component, Mrc1, a homologue of claspin. In response to DNA damage, Mec1 cooperates with the 53BP1 homologue Rad9. This dichotomy has led to the speculation that a replication fork stalled by nucleotide depletion adopts a structure distinct from one that is stalled by a lesion in the template (Alcasabas et al, 2001; Nielsen et al, 2013). However, the basis for such difference remains unclear.

Outside of S phase, ssDNA as a source of checkpoint activation can arise from nucleotide excision repair (NER) or from the resection of 5′-termini at DSBs or uncapped telomeres. In both situations, a 5′–3′ exonuclease, Exo1, contributes to Rad53 activation by widening NER gaps or processing DNA termini (Nakada et al, 2004; Dewar & Lydall, 2010; Giannattasio et al, 2010). At the same time, Exo1 is a downstream target of Rad53, which inhibits the nuclease by phosphorylation (Smolka et al, 2007; Morin et al, 2008). This results in a negative feedback that prevents excessive Exo1 activity. At stalled replication forks, Exo1 degrades abnormal structures and prevents fork reversal, but it does not contribute to damage signaling or replication restart and may even promote fork breakdown (Cotta-Ramusino et al, 2005; Segurado & Diffley, 2008). At collapsed replication forks, Exo1 activity is deemed to be mostly detrimental and is subject to checkpoint-mediated inhibition (Tsang et al, 2014).

1  Institute of Molecular Biology (IMB), Mainz, Germany
2  Cancer Research UK London Research Institute, Clare Hall Laboratories, Blanche Lane, South Mimms, UK
   *Corresponding author. Tel: +49 6131 3921490; E-mail: h.ulrich@imb-mainz.de
   †Present address: Springer Nature, London, UK
   ‡Present address: Frontier Research Institute for Interdisciplinary Sciences, Tohoku University, Aoba-ku, Sendai, Japan

As an alternative to persistent fork stalling, re-priming of DNA synthesis downstream of a lesion can give rise to daughter-strand gaps behind the replication fork. This has been studied most extensively in bacterial systems (Heller & Marians, 2006), but there is good evidence for a "skipping" of DNA damage-induced lesions in eukaryotic cells as well (Lopes *et al*, 2006; Elvers *et al*, 2011). Ultimately, however, cell proliferation requires complete genome replication, necessitating the activity of DNA damage bypass pathways to copy the damaged DNA (Friedberg, 2005; Ulrich, 2009). Importantly, these pathways, initiated by the ubiquitylation of the replication factor PCNA (Hoege *et al*, 2002) and involving either translesion synthesis by specialized, damage-tolerant polymerases or a recombination-like process named template switching, are not necessarily coupled to replication fork progression. They can be delayed without major effects on genome stability until bulk genome replication is completed (Ulrich, 2009; Daigaku *et al*, 2010; Karras & Jentsch, 2010), although an impact on the transmission of epigenetic information has been reported (Sarkies *et al*, 2010). Under these conditions, daughter-strand gaps accumulate and give rise to a damage response, accompanied by a cell cycle arrest in G2/M phase (Lopes *et al*, 2006; Callegari *et al*, 2010; Daigaku *et al*, 2010). When damage bypass is re-activated at that point, the pathway mediates the filling of these gaps in a postreplicative manner (Daigaku *et al*, 2010; Karras & Jentsch, 2010).

The significance of re-priming and daughter-strand gap formation for checkpoint signaling in *WT* cells is not well understood. A postreplication checkpoint that senses unreplicated DNA has been postulated (Callegari & Kelly, 2006), and Balint *et al* (2015) have described the assembly of a Mec1-activating complex distal to replication forks in response to DNA damage induced by the alkylating agent methyl methanesulfonate (MMS). However, the notion of postreplicative checkpoint activation contradicts the established concept of fork uncoupling, which invokes the stalled replication fork as the source of ssDNA that activates checkpoint signaling (Walter & Newport, 2000; Byun *et al*, 2005). In order to resolve this conflict, we made use of a genetic tool to delay damage bypass, thus causing a damage-dependent hyper-accumulation of daughter-strand gaps (Daigaku *et al*, 2010). In this setting, we identified an Exo1-dependent mechanism of Rad53 activation that in turn prevents erosion of gaps and an irreversible loss of viability largely attributable to the unrestrained activities of Exo1 and Pif1. Although reminiscent of the replication-independent action of Exo1 at DNA termini and NER gaps, this process required entry into S phase. Importantly, the same Exo1-dependent mechanism of Rad53 activation was observed in damage bypass-competent cells specifically during replication of damaged DNA, but not in response to nucleotide depletion or replisome problems. These findings explain the dichotomy between Mrc1- and Rad9-dependent Rad53 activation and suggest two distinct, spatially segregated mechanisms of how replication stress causes checkpoint activation: a fork-associated, Mrc1-dependent, Exo1-independent reaction in response to replisome-inherent problems and a gap-associated, Rad9- and Exo1-dependent process that predominates under conditions of template-induced polymerase stalling. We conclude that even in bypass-competent cells, regions of ssDNA left behind in the wake of replication forks and expanded by the action of processing factors such as Exo1, rather than stalled replication forks per se, constitute the predominant signal that leads to checkpoint activation in response to polymerase-stalling DNA lesions during S phase.

# Results

## Rad9-mediated checkpoint signaling is essential for damage resistance in the absence of damage bypass

In order to systematically explore the relationship between checkpoint activation and damage bypass, we depleted Rad18, the ubiquitin ligase responsible for initiating the pathway (Hoege *et al*, 2002), thus enforcing hyper-accumulation of daughter-strand gaps during replication over lesions (Daigaku *et al*, 2010; Karras & Jentsch, 2010). In order to avoid the accumulation of suppressors, we used a regulable allele, *Tet-RAD18*, which conveys a *rad18Δ*-like phenotype only in the presence of doxycycline (Daigaku *et al*, 2010). We monitored the effects of Rad18 loss on defined checkpoint mutants with respect to three different types of genotoxic stress: the methylating agent MMS, which elicits a damage response primarily during replication, 4-nitroquinoline oxide (4NQO), which forms bulky adducts that are perceived in a replication-independent manner, and HU, which causes replication fork stalling by means of nucleotide depletion without inducing lesions in the replication template. Depletion of Rad18 strongly sensitized the checkpoint mutants *mec1Δ*, *rad53Δ,* and *mrc1Δ rad9Δ* toward MMS and 4NQO, confirming the importance of checkpoint signaling in the absence of damage bypass (Fig 1A). Synergism was also observable with *rad9Δ* alone, but not with *mrc1Δ*, suggesting that the DNA damage checkpoint—as opposed to the replication checkpoint—was responsible for the effect. In support of this model, depletion of Rad18 only mildly enhanced the sensitivity of any of the strains toward moderate concentrations of HU, indicating that replication fork stalling per se is not particularly detrimental in the absence of Rad18. A *RAD18* deletion yielded comparable results (Fig EV1A). These findings imply a synergistic impact of damage bypass and specifically Rad9-dependent checkpoint signaling on the processing of DNA lesions.

## Rad9-mediated checkpoint signaling maintains damage bypass competence during S phase

Rad18 is a rate-limiting factor for PCNA ubiquitylation. Hence, the *Tet-RAD18* allele allows us to modulate the activation of the damage bypass pathway at will in the course of a cell cycle. In this manner, we had previously shown that synchronized cells, treated in the G1 phase with low doses of ultraviolet (UV) radiation in the absence of Rad18, replicate the bulk of their genomes, but stall in G2/M phase with an activated checkpoint due to the hyper-accumulation of daughter-strand gaps (Daigaku *et al*, 2010). *RAD18* re-expression at any time during or after genome replication allows them to recover, indicating that postreplicative gap filling can substitute for replication-associated damage bypass. We now used this approach to examine the mechanism of checkpoint activation under conditions of daughter-strand gap hyper-accumulation (Fig 1B): Alpha-factor (αF)-arrested G1 cells were UV-irradiated in the absence of Rad18 and subsequently released into S phase. *RAD18* expression was then induced either immediately upon release, in mid-S phase, or after

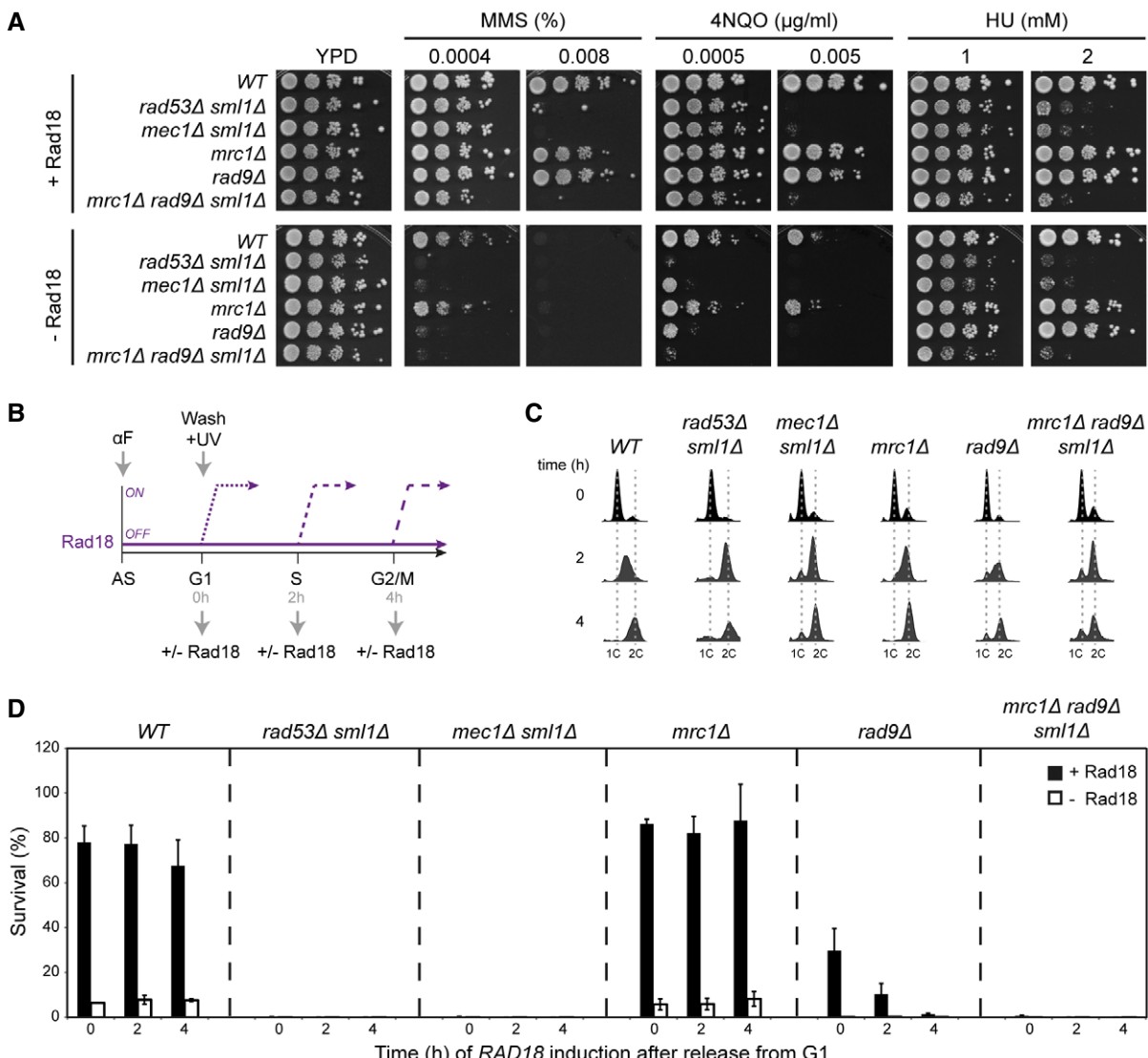

**Figure 1. Contribution of checkpoint factors to DNA damage bypass.**

A  DNA damage sensitivities of *Tet-RAD18* strains carrying the indicated gene deletions, determined by growth assays in the presence (top) or absence (bottom) of Rad18.

B  Experimental scheme for measuring recovery of viability after UV irradiation (20 J/m²) upon *Tet-RAD18* induction at the indicated times after release into S phase (AS: asynchronous; αF: alpha-factor). For details, see Materials and Methods.

C  Cell cycle profiles of the indicated strains at the time of plating.

D  Survival of the indicated strains, relative to unirradiated controls. Error bars indicate SD derived from three independent experiments.

cells had reached G2/M phase (Fig 1C), and survival was determined by plating of aliquots. As previously reported, checkpoint-proficient (*WT*) cells recovered viability independently of the timing of *RAD18* induction (Daigaku *et al*, 2010). In contrast, *mec1Δ* and *rad53Δ* mutants were completely unable to recover (Fig 1D). In the case of *mec1Δ*, the defect might be ascribed to a direct participation of the kinase in translesion synthesis by phosphorylation of Rev1 (Pages *et al*, 2009), but for *rad53Δ* this does not apply. Here, the defect was neither due to a failure to re-express *RAD18* (Fig EV1B) nor caused by the UV sensitivity conferred by the *rad53Δ* mutation itself, as viability remained consistently higher when the assay was

performed in the continuous presence of Rad18 (Fig EV1C and D). Hence, in *rad53Δ* cells even a temporary absence of Rad18 appears to cause a complete and irreversible loss of the capacity to productively use damage bypass. The ability to recover by *RAD18* induction depended on the kinase activity of Rad53, as a catalytically deficient mutant, *rad53-K227A*, did not regain viability (Fig EV1E). As Rad53 can be activated either via the Rad9-dependent damage checkpoint or the Mrc1-dependent replication checkpoint (Pardo *et al*, 2017), we examined recovery of viability in *rad9Δ* and *mrc1Δ* mutants. As shown in Fig 1D, the *mrc1Δ* mutant fully recovered upon *RAD18* re-expression, whereas deletion of *RAD9* caused a

significant loss of viability that grew successively more severe with a prolonged delay of *RAD18* induction. In contrast, deletion of *RAD9* or *MRC1* in the presence of *RAD18* had little effect on viability (Fig EV1D). This observation suggests that the Rad9-mediated damage checkpoint, rather than the Mrc1-dependent replication checkpoint, is essential to maintain damage bypass competence as cells progress through S phase. An *mrc1Δ rad9Δ* double mutant phenocopied *rad53Δ* mutants (Figs 1D and EV1F), indicating that Mrc1-mediated checkpoint signaling may partially compensate for the loss of Rad9 during early S phase. Consistent with this model, Rad53 phosphorylation was severely reduced upon *RAD9* deletion, but completely abolished in the *mrc1Δ rad9Δ* double mutant (Fig EV1G). Hence, our findings suggest that Rad9-mediated activation of the Rad53 kinase becomes essential when damage bypass is delayed.

## Delay of damage bypass in *rad53Δ* mutants causes elevated homologous recombination and catastrophic chromosome fragmentation

We next sought to elucidate how checkpoint mutants lost viability upon inhibition of damage bypass. Using pulsed-field gel electrophoresis (PFGE), we found that UV-irradiated *rad53Δ* cells grown in the absence of Rad18 failed to restore the pattern of intact chromosomes indicative of successful completion of genome replication (Fig 2A). Instead, we observed substantial chromosome fragmentation in the course of S phase (Figs 2A and EV2A). This was also observed in *rad9Δ*, but not in *mrc1Δ* cells (Fig EV2B). Consistent with the fragmentation pattern, *rad53Δ* cells in the absence of Rad18 accumulated strongly elevated numbers of recombination foci and exhibited aberrant chromosome segregation patterns (Figs 2B and C, and EV2C and D). Importantly, both chromosome breaks and hyper-accumulation of Rad52$^{YFP}$ foci were only observed in the absence of Rad18. From these observations, we conclude that when damage bypass fails, Rad9-mediated checkpoint signaling is essential to prevent massive chromosome fragmentation during S phase and—likely as a consequence of this—an elevated frequency of aberrant or failed divisions. Similar defects had been reported in *rad18Δ* cells in response to low doses of chronic damage (Hishida *et al*, 2009).

## Rad53 is required during S phase, but not for postreplicative gap filling

The failure of *rad53Δ* mutants to reactivate damage bypass might be due to a direct requirement of Rad53 for the filling of daughter-strand gaps. However, the successive loss of chromosome integrity in the course of S phase suggested an essential function of checkpoint signaling already at the stage where the gaps emerge. In order to distinguish between these models, we used a previously characterized allele, *rad53$^{AID*−9myc}$*, that encodes the kinase as a fusion with an auxin-inducible degron (Morawska & Ulrich, 2013). This allows depletion of the protein within < 1 h and confers a *rad53Δ*-like phenotype in the presence, but *WT* behavior in the absence of auxin (Morawska & Ulrich, 2013; Appendix Fig S1A). With this allele, we re-examined the recovery of viability under conditions where Rad53$^{AID*−9myc}$ was depleted either prior to the start of S phase (Fig 3A) or after passage through S phase, but before

reactivation of Rad18 (Fig 3B). As expected, when Rad53$^{AID*−9myc}$ was removed prior to UV treatment, recovery was strongly compromised (Fig 3A). The defect was not as severe as in a *rad53Δ* strain, but this may have been due to residual protein even in the presence of auxin. Recovery was normal in the absence of auxin (Appendix Fig S1B), indicating that the failure to restore viability was indeed a consequence of the degradation of Rad53, and the AID*-tagged protein was functional under stabilizing conditions. Degradation of Rad53$^{AID*−9myc}$ after completion of S phase had no detrimental effect on viability, suggesting that Rad53 function is dispensable for damage bypass in G2/M (Fig 3B and Appendix Fig S1C and D).

We then set up an experiment where Rad53$^{AID*−9myc}$ was temporarily degraded before release from G1 but re-expressed together with Rad18 at different times during the cell cycle (Fig 3C, -Aux). When *rad53$^{AID*−9myc}$* was re-expressed before entry into S phase (0 h), cells recovered viability. However, if re-expression was postponed to mid-S (2 h) or G2 phase (4 h), loss of viability became irreversible, even though recovery of Rad53$^{AID*−9myc}$ protein levels resulted in a restoration of checkpoint signaling (assessed by the upregulation of the ribonucleotide reductase subunit, Rnr4; Appendix Fig S1E). Taken together, these data indicate that a transient loss of Rad53 during replication is sufficient to irreversibly prevent productive damage bypass.

## Rad53-mediated inhibition of Exo1 and Pif1 maintains bypass competence during S phase

In order to identify the mechanism(s) by which Rad53 maintains damage bypass competence, we systematically examined possible contributions of Rad53's downstream targets. We found that upregulation of dNTP levels was required for efficient damage bypass, but not sufficient to restore viability in *rad53Δ* cells (Fig EV3A). Abolishing the suppression of late origin firing (Lopez-Mosqueda *et al*, 2010; Zegerman & Diffley, 2010) in a Rad53-proficient background strongly accelerated progression through S phase, but did not interfere with viability (Fig EV3B). *Vice versa*, delay of mitosis by nocodazole treatment did not rescue viability in the absence of Rad53 (Fig EV3C). We were also able to exclude a contribution of Rad53-induced gene expression controlled by the transcriptional co-repressor Nrm1 (de Bruin *et al*, 2006; Travesa *et al*, 2012; Fig EV3D) and an influence of histone gene dosage, which had also been shown to affect the damage sensitivity of *rad53Δ* mutants (Gunjan & Verreault, 2003; Fig EV3E). Finally, in order to assess whether elevated homologous recombination was the underlying cause of the problems or rather a reflection of (unsuccessful) attempts at repair, we analyzed the effects of various mutants defective in distinct stages of homologous recombination, such as *mre11Δ, rad55Δ, mms4Δ, slx4Δ, yen1Δ, sgs1Δ,* and *srs2Δ*. However, none of them restored Rad18-mediated survival in a *rad53Δ* background (Fig EV4).

Having excluded recombination as a source of genome instability, we considered pathological expansion of ssDNA as a cause of the observed chromosome damage. At HU-stalled replication forks, the excessive formation of ssDNA and fork breakdown that is observable in checkpoint mutants is mainly promoted by Exo1, Pif1, and Rrm3 (Cotta-Ramusino *et al*, 2005; Rossi *et al*, 2015). The latter two also contribute to unperturbed replication by resolution of

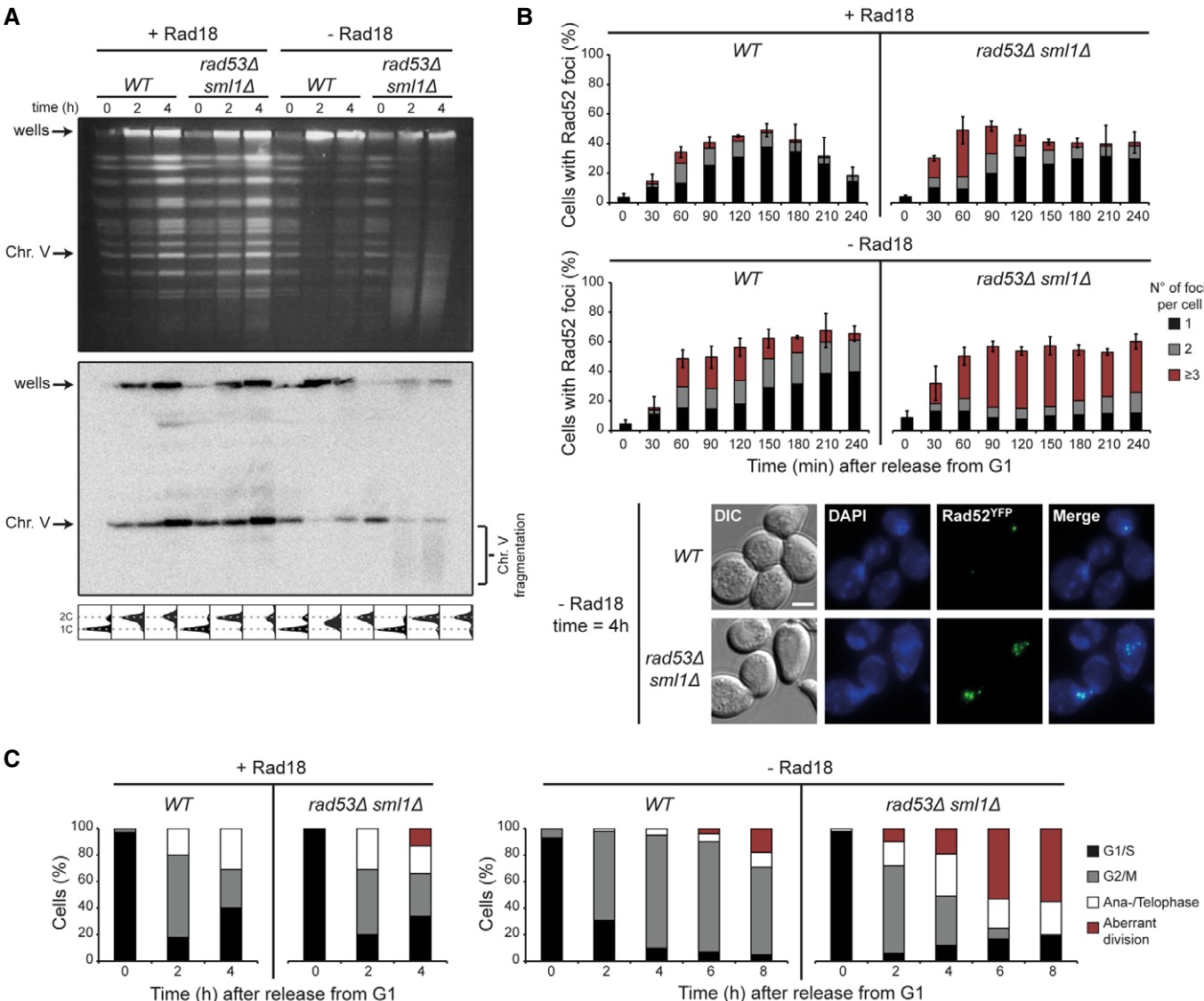

**Figure 2.  Delay of damage bypass in *rad53Δ* mutants causes chromosome fragmentation, excessive recombination, and aberrant division.**

*WT* and *rad53Δ* cells were grown in the presence or absence of Rad18, synchronized in G1, UV-irradiated, and released into S phase.

A   Yeast chromosomes, analyzed by pulsed-field gel electrophoresis and ethidium bromide staining (top) or Southern blotting for chromosome V (middle). Replication intermediates accumulate in the wells. Cell cycle profiles are shown below the respective strains.

B   Quantification of Rad52[YFP] recombination foci and representative images. Error bars indicate SD derived from three independent experiments. Scale bar = 5 μm.

C   Analysis of mitotic aberrations. Cells were classified into cell cycle stages according to spindle morphology (see Fig EV2C and D for examples).

problematic sequences and DNA–protein complexes, respectively (Ivessa *et al*, 2003; Paeschke *et al*, 2011; Sabouri *et al*, 2012), and all of them are inhibited by Rad53-mediated phosphorylation (Smolka *et al*, 2007; Morin *et al*, 2008; Rossi *et al*, 2015). Consistent with a contribution to daughter-strand gap erosion, deletion of *EXO1* or *PIF1* significantly improved viability of *rad53Δ* mutants in our recovery assay (Fig 4A), while they had little influence in a checkpoint-proficient background (Fig EV5A). A combination of *exo1Δ* and *pif1Δ* was even more effective, although it should be noted that recovery was still far from complete. In contrast, *rrm3Δ* did not affect recovery alone or in combination with *exo1Δ* (Fig 4A). Accordingly, we found Exo1 and Pif1 to be phosphorylated in a

Rad53-dependent manner after UV irradiation and release into S phase (Fig 4B). Phosphorylation of Exo1 followed the pattern observed for Rad53 itself under these conditions, that is, it was mediated mainly via Rad9, with Mrc1 playing a backup role only in the absence of Rad9 (Fig EV5B).

If Rad53's predominant role in maintaining damage bypass competence were indeed the suppression of Exo1 and Pif1, a Rad53-insensitive Exo1 should confer a recovery defect comparable to a checkpoint mutant in our assay. Consistent with this prediction, replacement of *EXO1* by a previously described allele, *exo1-SA*, where the four major phosphorylation sites were mutated to alanine (Morin *et al*, 2008; Doerfler & Schmidt, 2014), considerably reduced

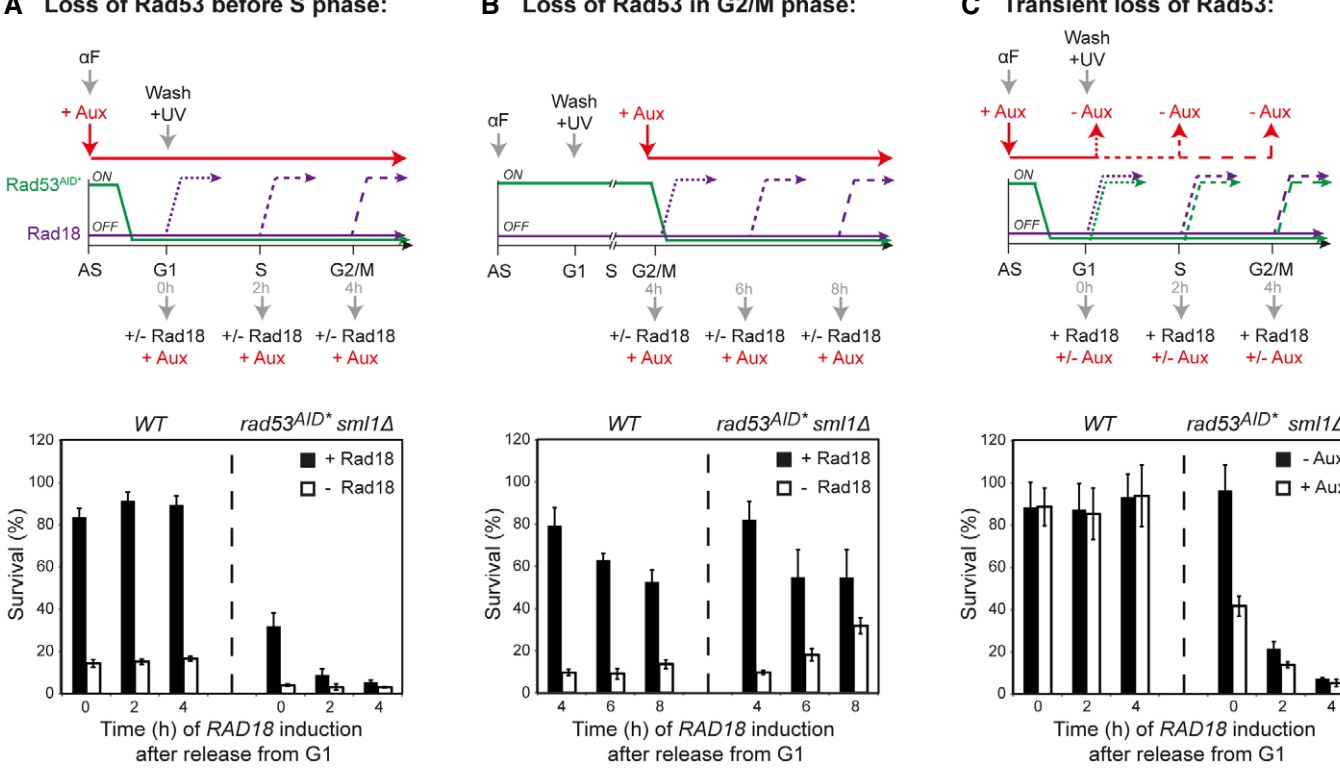

**Figure 3.  Rad53 is required during the S phase that precedes DNA damage bypass.**

A  Loss of Rad53 before S phase: recovery assays upon *RAD18* induction were performed as described in Fig 1B, but Rad53[AID*−9myc] degradation was induced by adding auxin during synchronization.

B  Loss of Rad53 in G2/M phase: assays were performed as above, but Rad53[AID*−9myc] degradation was induced 4 h after release into S phase.

C  Transient loss of Rad53: recovery was measured after Rad53[AID*−9myc] degradation during synchronization and re-expression together with *RAD18* at the indicated times during the cell cycle.

Data information: (A–C) Error bars indicate SD derived from at least three independent experiments.

the capacity to restore viability (Fig 4C). This effect was not primarily a consequence of a general damage sensitivity of the *exo1-SA* mutant because in the presence of Rad18, survival was similar to *WT* (Fig EV5C). The defect conferred by *exo1-SA* was dependent on the protein's nuclease activity, since its inactivation (in *exo1-SA-ND*) restored *WT* levels of survival (Fig 4C). Moreover, the Exo1-SA protein had a dominant effect (Fig 4D), and combination with *rad53Δ* did not exacerbate the situation, indicating an epistatic relationship (Fig 4E). However, the phenotype of the *exo1-SA* mutant was significantly milder than that of *rad53Δ*. While this may imply additional Rad53 targets involved in controlling the stability of postreplicative gaps, for example, Pif1, part of the effect could also be due to a phosphorylation of Exo1 at additional sites not covered by the *exo1-SA* allele (Morin *et al*, 2008). Indeed, analysis of the mutant protein revealed residual phosphorylation (Fig EV5D).

The notion that Rad53 restricts Exo1 and Pif1 activity can explain why checkpoint signaling is essential during replication when damage bypass is delayed, but it does not account for the observation that daughter-strand gaps remain bypass-competent for extended periods if Rad53 is depleted after cells have reached G2/M phase (Fig 3B, 6 h and 8 h). When we monitored the state of Exo1 tagged with an HA-epitope in *rad53[AID*−9myc]*, we noted

that—independent of the presence or absence of Rad53—protein levels rapidly declined at the end of S phase (Fig EV5E). This was neither due to our experimental set-up nor a consequence of the HA-tag, as an Exo1[9myc] protein exhibited similar, damage-independent fluctuation along the cell cycle in an otherwise unmodified strain, with protein levels peaking in S phase and declining in G2/M (Fig EV5F). The same pattern was observed for Pif1 (Fig EV5G). Thus, the cell cycle regulation of Exo1 and Pif1 apparently obviates the need for Rad53-mediated inhibition in G2/M, and inactivation of Rad53-dependent damage signaling at this stage can therefore no longer interfere with productive damage bypass (Fig 3B). In addition, Exo1 dephosphorylation after degradation of Rad53 proceeded very slowly (Fig EV5E), which might contribute to a sustained repression of any Exo1 activity remaining in G2/M.

### Damage signaling during S phase requires Exo1 activity at daughter-strand gaps

At DSBs and NER gaps outside of S phase, Exo1 is subject to a Rad53-dependent feedback regulation where Exo1 itself generates the checkpoint signal that ultimately restricts its own activity (Nakada *et al*, 2004; Dewar & Lydall, 2010; Giannattasio *et al*,

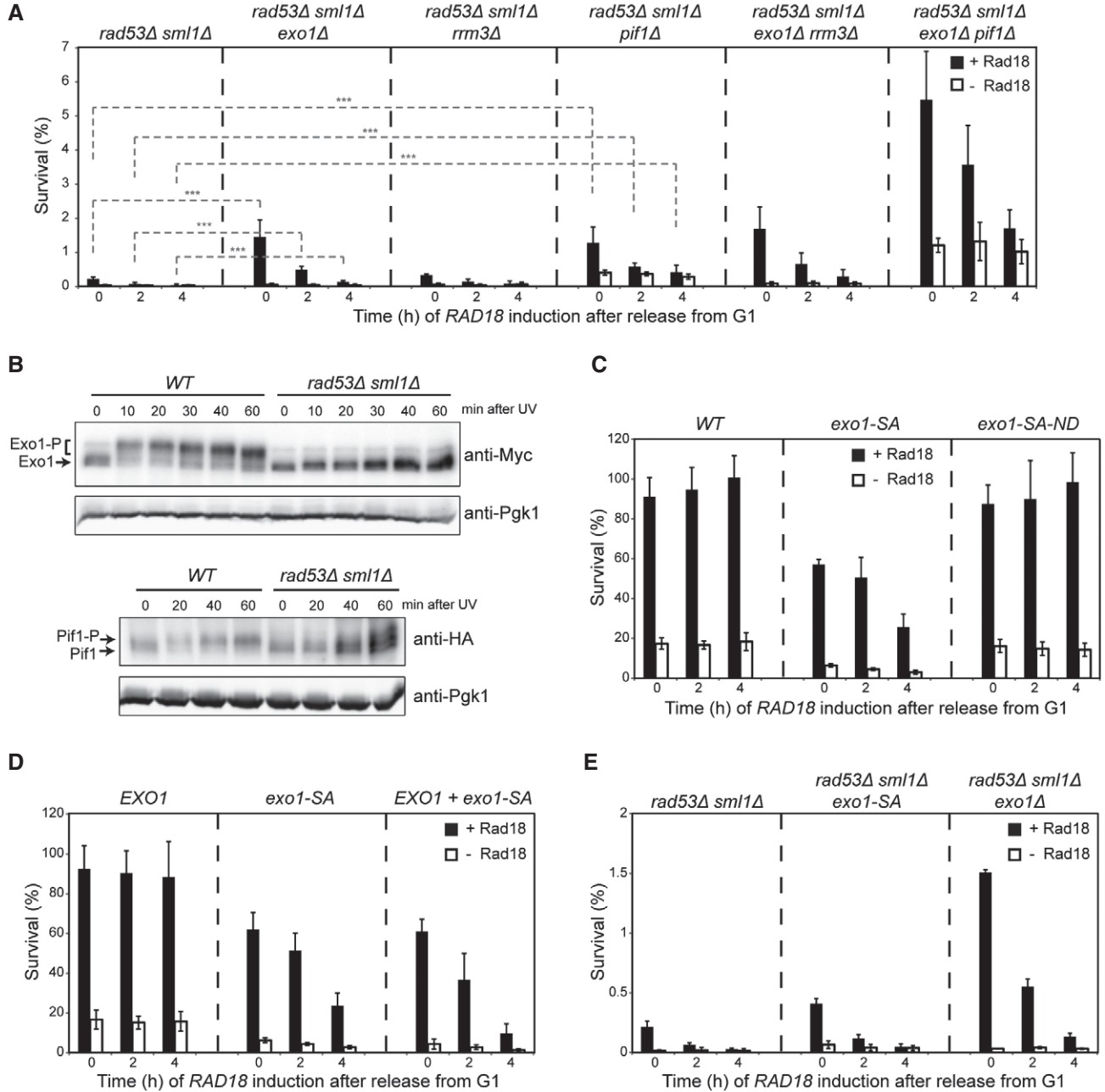

**Figure 4. Checkpoint-mediated inhibition of Exo1 and Pif1 activity is required to maintain bypass competence.**

A   Recovery of viability upon *RAD18* induction in the indicated strains, measured as described in Fig 1B (***P < 0.001).

B   Exo1 and Pif1 phosphorylation in the indicated strains, released into S phase after UV irradiation in the absence of Rad18. Exo1[9myc] and Pif1[6HA] were detected by Western blotting. Pgk1 served as loading control.

C   Recovery of viability in *exo1* mutants (SA: dephospho-mimicking; ND: nuclease-dead).

D   Recovery of viability in strains harboring wild-type *EXO1*, *exo1-SA* mutant or both alleles (*EXO1 + exo1-SA*).

E   Recovery of viability in *rad53Δ exo1-SA*.

Data information: (A, C–E) Error bars indicate SD derived from at least three independent experiments. Significance in panel (A) was calculated by the Student's *t*-test. Source data are available online for this figure.

2010). If such phenomenon also applied at daughter-strand gaps during S phase, deletion of *EXO1* should interfere with Rad53 activation in our assay. Indeed, when cells were synchronized, UV-irradiated, and released into S phase in the absence of Rad18,

phosphorylation of Rad53 was significantly delayed in *exo1Δ* cells, resulting in faster cell cycle progression (Fig 5A). Under these conditions, Mre11 can apparently compensate for the lack of Exo1 to some extent, as *mre11Δ* cells activated Rad53 like *WT*, but the

*exo1Δ mre11Δ* double mutant exhibited a further delay in Rad53 phosphorylation and a severely accelerated S phase. Compared to Exo1, Pif1 contributed much less to checkpoint activation in S phase (Fig 5B). These observations strongly suggest that Exo1-mediated resection generates the signal for timely activation of the checkpoint at daughter-strand gaps accumulating in the absence of Rad18.

The observed relationship between Exo1 and Rad53 at daughter-strand gaps prompted us to examine the mechanism of checkpoint activation in damage bypass-competent cells. As shown in Fig 5C, despite an overall reduced signal, Rad53 activation after UV irradiation followed the same pattern in the presence of Rad18 as in its absence, with a strong dependence on Exo1 and a minor contribution of Mre11 at later time points. Notably, such relationship did not apply when replication fork stalling was induced by HU treatment, which presumably impinges on replisome progression without generating lesions (Fig 5D). In this situation, Rad53 was efficiently activated in an Exo1-independent, partially Mre11-dependent manner, consistent with previous reports (Nakada *et al*, 2004). We therefore asked under which conditions of replication stress Exo1 would contribute to Rad53 activation. MMS, which causes polymerase stalling due to lesions in the replication template, induced Rad53 phosphorylation in an Exo1-dependent manner, as observed with UV (Fig 5E, Appendix Fig S2). As an alternative, template-independent source of replication stress, we depleted polymerase α (Pol1) in G1-arrested cells by means of an auxin-inducible degron tag (Morawska & Ulrich, 2013). Subsequent release into S phase caused a strong spontaneous checkpoint response in $pol1^{AID*-9myc}$ cells, most likely because of a problem with initiating lagging strand synthesis (Fig 5F). In this situation, Rad53 phosphorylation was independent of Exo1, as observed in response to HU. A replication-independent checkpoint activation, for example, by NER gaps, was ruled out by the lack of UV- or MMS-induced Rad53 phosphorylation in cells maintained in G1 phase (Fig 5G and H).

### DNA lesions and replisome problems activate checkpoint signaling via distinct structures

Our observations imply that replication-dependent, lesion-induced Rad53 activation follows a similar Exo1-mediated mechanism as the replication-independent process at NER gaps. This strongly suggests that even in damage bypass-competent cells the signal that activates Rad53 in response to DNA damage emanates from daughter-strand

gaps and not from stalled replication forks. In order to obtain direct evidence for this model, we visualized the distribution of ssDNA relative to regions of newly replicated DNA on fibers isolated from early S phase cells under conditions of replication stress and calculated the total percentage of ssDNA along each replication tract (Fig 6A and B). Consistent with an accumulation of daughter-strand gaps, we found elevated levels of ssDNA along the lengths of many replication tracts on DNA fibers from UV- or MMS-treated cells. This pattern was strikingly different when fork stalling was induced by nucleotide depletion, irrespective of HU concentration (Fig 6A and B, Appendix Fig S3A–C). In a second experiment, we compared the MMS-induced enrichment of ssDNA within replicated regions to the corresponding enrichment in the EdU-negative, that is, unreplicated areas in the same set of fibers. Outside the replication tracts, MMS treatment caused an approximately twofold increase in the total percentage of ssDNA, largely reflecting a higher density of tracts (Fig 6C and D). Within the EdU-labeled regions, tract density was also twofold higher, but the total amount of ssDNA was enriched by almost sixfold. Hence, the accumulation of ssDNA in the newly replicated DNA is mainly attributable to an increase in tract length (Fig 6C and D). These results suggest that the majority of ssDNA within replicated areas corresponds to postreplicative daughter-strand gaps, and only a minority may have resulted from replication-independent processes such as an expansion of NER gaps.

In an *exo1Δ* mutant, the fraction of ssDNA upon MMS treatment was more than twofold reduced compared to *WT* cells, while the number of tracts per unit length was similar, indicating that Exo1 indeed influences the length of the tracts, but not their incidence (Fig 6C and D). Taken together, our data reveal two distinct mechanisms by which replication stress can generate a checkpoint signal: a fork-associated, Exo1-independent mechanism that responds to replisome problems or nucleotide depletion, and an Exo1-and Rad9-dependent process that is induced by lesions in the replication template and emanates from postreplicative daughter-strand gaps.

## Discussion

### Lesion-induced replication stress is perceived behind the fork

Numerous studies have addressed the question of how replication stress is sensed and converted to a global checkpoint signal (Branzei

---

**Figure 5.  Exo1 is required for robust checkpoint activation in response to polymerase-blocking lesions but not replisome-induced fork stalling.**

A  Rad53 phosphorylation in the indicated strains, synchronized in G1, UV-irradiated (20 J/m²), and released into S phase in the absence of Rad18. Pgk1 served as loading control. Cell cycle profiles are shown below the blots.

B  Rad53 phosphorylation in the indicated strains, treated as in panel (A).

C  Rad53 phosphorylation in the indicated strains, treated as in panel (A) but grown in the presence of Rad18.

D  Rad53 phosphorylation in the indicated strains, synchronized in G1, and released into S phase in medium containing HU (120 mM).

E  Rad53 phosphorylation in the indicated strains, synchronized in G1, treated with MMS (0.08%) for 30 min, and released into S phase.

F  Rad53 phosphorylation in the indicated strains, synchronized in G1, and released into S phase. Auxin was added 30 min prior to release for induction of Pol1$^{AID*-9myc}$ degradation.

G  Rad53 phosphorylation in *WT* cells, synchronized in G1, undamaged, or UV-irradiated (20 J/m²) and either maintained in G1 or released into S phase for 20 min.

H  Rad53 phosphorylation in *WT* cells, synchronized in G1, untreated or treated with 0.08% MMS for 30 min, and either maintained in G1 or released into S phase for 10 min.

Source data are available online for this figure.

**A** **UV damage, - Rad18:**

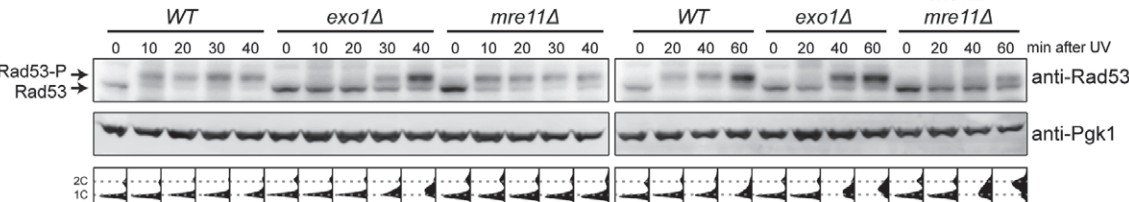

**B** **UV damage, - Rad18:**

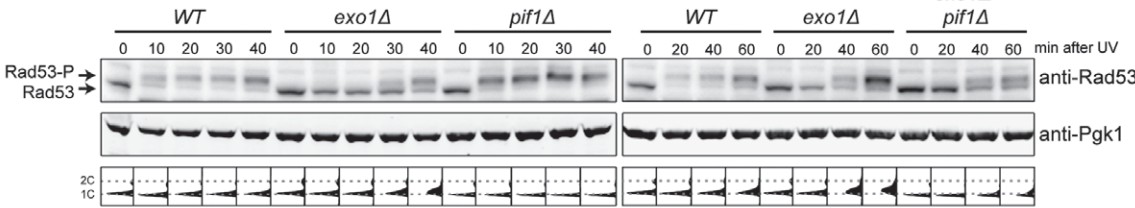

**C** **UV damage, + Rad18:**

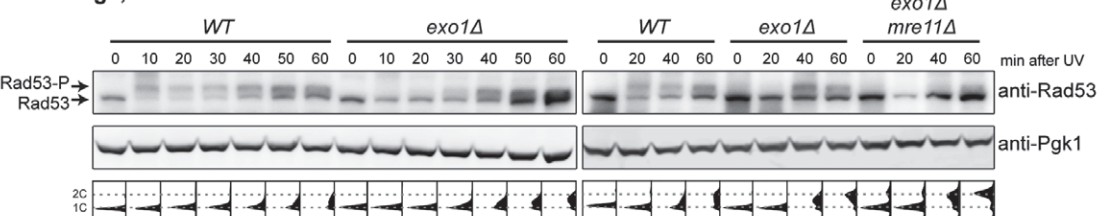

**D** **HU treatment, + Rad18:**

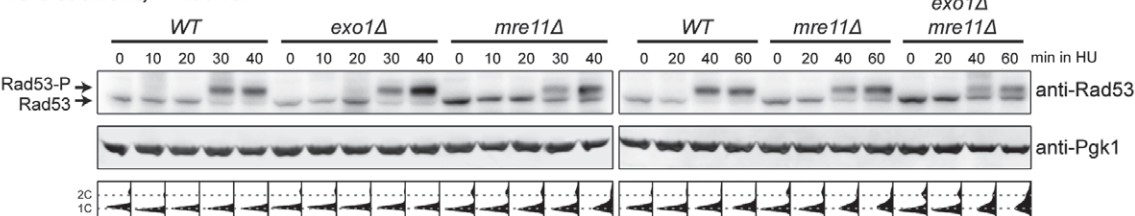

**E** **MMS damage, + Rad18:**

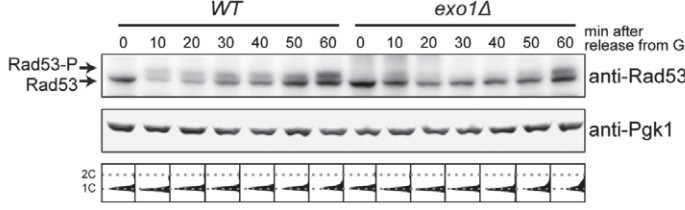

**G**

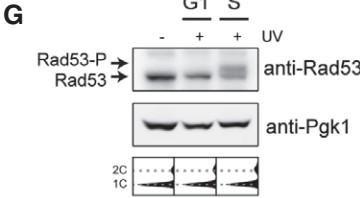

**F** **Polα depletion, + Rad18:**

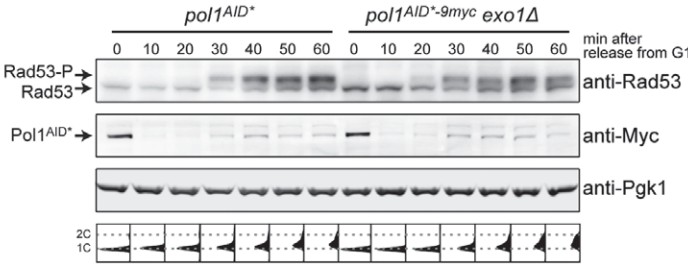

**H**

**Figure 5.**

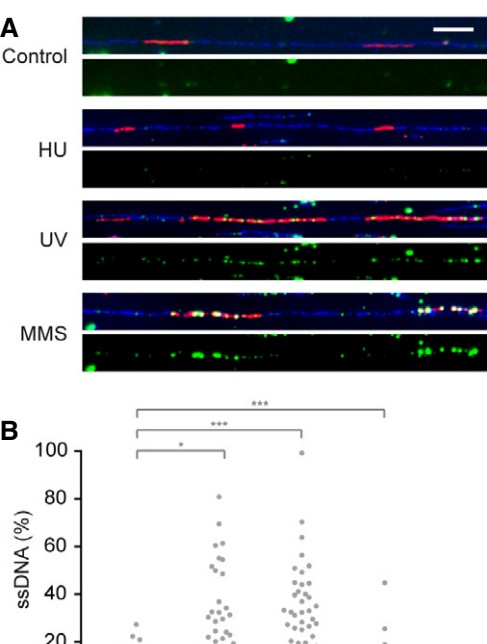

**Figure 6.  ssDNA accumulates within tracts of newly synthesized DNA in response to polymerase-blocking lesions.**

A   DNA fibers, prepared by combing of genomic DNA isolated from cells synchronized in G1 and released into S phase in the presence of EdU (added 15 min before release). Cells were harvested 20 min after release for control; 30 min after UV irradiation (20 J/m²); 30 min after treatment with 0.04% MMS for 30 min prior to release; and 60 min after release into 120 mM HU. Fibers were stained with YOYO-1 for total DNA (blue). EdU incorporation was visualized by a click reaction with Alexa Fluor 647 (red), and ssDNA was detected by means of an antibody (green). Scale bar = 10 kbp.

B   Quantification of the fraction of ssDNA within newly replicated DNA, determined for individual EdU-stained tracts by measuring total tract length and total length of ssDNA within that tract. Evidence for EdU-stained regions representing replication tracts is shown in Appendix Fig S3D and E. Number of replication tracts analyzed: Control = 186; UV = 204; MMS = 168; HU = 198.

C   Quantification of the fraction of ssDNA within or outside of EdU-stained replication tracts derived from *WT* or *exo1Δ* cells, determined as above. G1-arrested cells were incubated with or without 0.02% MMS for 30 min and released into EdU for 30 min. Number of replication tracts analyzed: *WT* control = 63; *exo1Δ* control = 151; *WT* MMS = 124; *exo1Δ* MMS = 122. Number of EdU-negative tracts analyzed: *WT* control = 96; *WT* MMS = 171.

D   Density of ssDNA tracts within or outside of individual replication tracts from the experiment shown in panel (C), calculated by dividing the number of ssDNA tracts by the length (in kb) of the corresponding EdU-stained or EdU-negative region.

Data information: (B–D) Significance was calculated by the Mann–Whitney test (ns: not significant; *P < 0.05; **P < 0.01; ***P < 0.001). Black bar = mean.

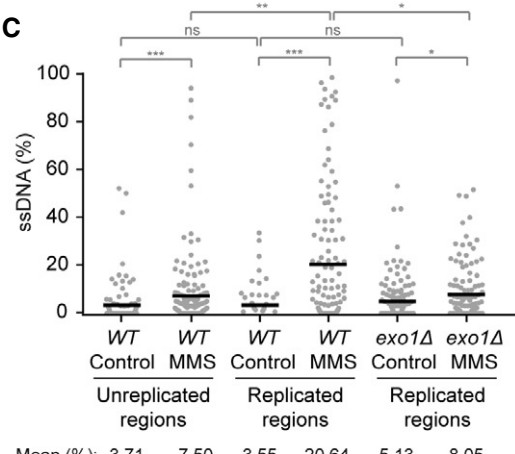

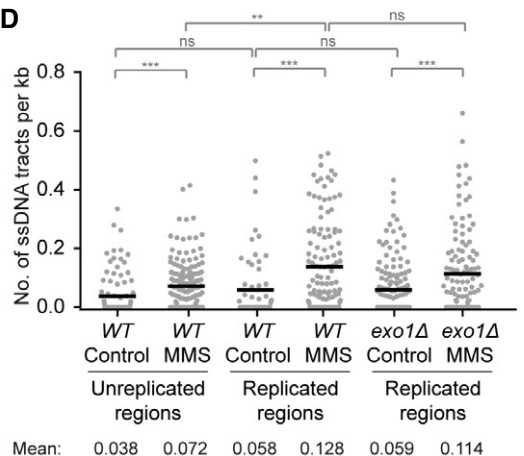

& Foiani, 2009). The general concept that has emerged from these studies postulates that checkpoint kinases are activated by extended regions of ssDNA arising at stalled forks from an uncoupling between replicative helicase and DNA polymerase movement or between leading and lagging strand synthesis, and that checkpoint signaling maintains genome stability by stabilizing such structures (Byun *et al*, 2005; Branzei & Foiani, 2009). A number of reports have highlighted important differences in the mechanism by which different agents activate replication-specific checkpoint signaling. For example, Segurado and Diffley (Segurado & Diffley, 2008) have described two ways of replication fork stabilization by Rad53: an Exo1-dependent pathway that mainly reacts to MMS, UV or ionizing radiation damage, and an Exo1-independent mechanism that is more important after exposure to HU. Similarly, Nielsen *et al* (2013) have proposed that an HU-stalled fork adopts a structure distinct from one that is stalled by MMS, consistent with the notion that the replisome component Mrc1 dominates HU-induced checkpoint signaling, whereas the response to MMS depends on Rad9 (Alcasabas *et al*, 2001; Crabbe *et al*, 2010; Nielsen *et al*, 2013). Yet, the stalled replication fork as the origin of the checkpoint signal in response to replication stress has not been called into question.

We now suggest that the two pathways of sensing replication stress not only differ in their dependence on Mrc1 versus Rad9, but also with respect to their origin (Fig 7). In line with previous models, the HU-responsive, Mrc1-dependent pathway primarily monitors the state of the replisome and is thus closely associated with the fork structure itself (Fig 7A). However, we propose that the Rad9-dependent pathway that mediates the response to lesions in the replication template originates not from the fork, but rather from daughter-strand gaps, left behind after passage of the replisome and

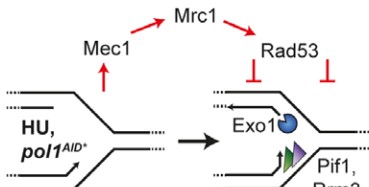

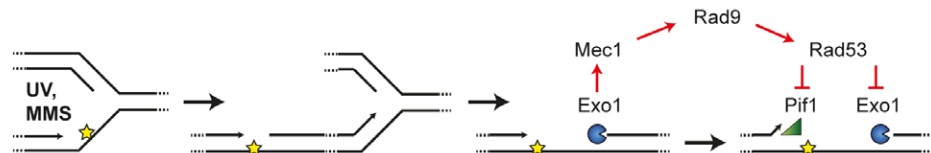

**Figure 7.  Model for differential checkpoint activation in response to replisome problems and DNA lesions.**

A  Upon replisome stalling—after HU treatment or Pol1 degradation—an excess of ssDNA at arrested forks activates Rad53 via Mrc1, which prevents replication fork breakdown by inhibition of Exo1, Pif1 and Rrm3 activities.

B  Replication through damaged DNA—after UV irradiation or MMS treatment—generates daughter-strand gaps behind replication forks due to re-priming events, thereby reducing the exposure of ssDNA at forks. Exo1-mediated expansion of daughter-strand gaps is then required for robust Rad9-dependent Rad53 activation, which in turn leads to Exo1 and Pif1 inhibition by phosphorylation.

expanded by the action of Exo1 in order to generate a sufficient checkpoint signal (Fig 7B). The notion that an establishment of active replication forks is required for Rad53 activation in response to damaging agents such as MMS (Tercero *et al*, 2003) does not contradict this concept, as passage of a replication fork would be a prerequisite for the accumulation of gaps. Moreover, although our model originates from observations made under conditions where a delay of damage bypass exacerbates the consequences of checkpoint failure, we found that the same mechanism applies in damage bypass-competent cells. Finally, the idea of Rad9-dependent checkpoint activation by regions of ssDNA embedded in a chromatinized environment at some distance from the replication fork is consistent with the mechanism of Rad9 recruitment by binding to Lys79-methylated histone H3 (Wysocki *et al*, 2005). Brown and coworkers have recently described an assembly of a signaling complex in discrete domains behind replication forks that leads to activation of Mec1 in response to MMS (Balint *et al*, 2015), and our recent observation that ubiquitylation of histone H2B at Lys123, a prerequisite for efficient H3 methylation, contributes to postreplicative damage bypass, also supports this notion (Hung *et al*, 2017). Thus, our results complement published data and offer a satisfying mechanistic explanation for the differences between replisome- and template-induced checkpoint activation.

### Re-priming as a means to resolve replication fork uncoupling

The model posits that lesions in the template strand do not pose a significant block to the overall progression of the replisome, but rather induce downstream re-priming of DNA synthesis, thus giving rise to daughter-strand gaps. Indeed, re-priming has been well documented in various experimental systems (Heller & Marians, 2006; Lopes *et al*, 2006; Callegari *et al*, 2010; Hashimoto *et al*, 2010; Elvers *et al*, 2011). Thus, if damage signaling were responsive only to fork-associated ssDNA, efficient re-priming should work against activation of the Mrc1-mediated checkpoint, which requires a critical number of arrested forks (Shimada *et al*, 2002; Tercero *et al*, 2003). Gap-associated checkpoint activation via Rad9 would overcome this problem. Our model is therefore at odds with the established concept of replication fork uncoupling that has been proposed to explain the origin of the ssDNA required

for checkpoint activation in response to both polymerase inhibition and UV- or cis-platinum-induced lesions (Walter & Newport, 2000; Byun *et al*, 2005). It has to be noted that that model, derived from experiments in *X. laevis* egg extracts, relied on the detection of ssDNA in a plasmid template as a readout without specifying whether the ssDNA accumulated at or behind the fork. Indeed, in a similar set-up Hashimoto *et al* (2010) readily observed daughter-strand gaps, but very little strand uncoupling in replicating MMS-damaged sperm DNA by means of electron microscopy. We would like to point out, however, that at severe damage load, re-priming may eventually become inefficient due to an extremely high density of lesions, such that checkpoint signaling could then become fork-associated even in response to UV or MMS. In support of this view, even moderate doses of MMS have been reported to inhibit late origin firing in an Mrc1-dependent manner, suggesting that the replisome is affected to some extent by damaged templates, even though this may not compromise viability (Hang *et al*, 2015).

### Exo1 contributes to checkpoint activation in response to damage-induced replication stress

Exo1 is well known for its contribution to checkpoint activation by mediating resection of 5′-termini or widening of NER gaps outside of S phase (Dewar & Lydall, 2010; Giannattasio *et al*, 2010). In both cases, Rad53 limits Exo1 activity by a negative feedback control. We now show that a comparable situation applies to UV- or MMS-induced checkpoint signaling in S phase cells, raising the question whether the phenomenon observed here is actually replication-dependent. However, as also reported by others (Tercero *et al*, 2003), at the low doses of MMS applied here, Rad53 phosphorylation strictly depended on the initiation of S phase, thus ruling out a replication-independent mechanism. The same had previously been shown for both *WT* and *rad14Δ* cells in response to UV, arguing against an involvement of NER (Neecke *et al*, 1999). Finally, we observed DSBs only under conditions where both damage bypass and checkpoint signaling were compromised (Fig 2A). Hence, we conclude that the features giving rise to damage signaling in our system are neither DSBs nor NER gaps, but replication-dependent structures.

Yet, the function of Exo1 observed here appears to be distinct from its fork-associated roles: at HU-stalled or collapsed forks, Exo1 does not contribute to damage signaling or replication restart and may even cause fork breakdown (Cotta-Ramusino et al, 2005; Segurado & Diffley, 2008; Tsang et al, 2014). In contrast, in response to UV or MMS damage, Exo1 mediates Rad53 activation. Moreover, previous reports indicate that DNA damage bypass by template switching initiates from internal tracts of ssDNA rather than free termini, and Exo1 itself participates in this process, likely by expanding them in preparation for strand invasion (Vanoli et al, 2010; Karras et al, 2013; Giannattasio et al, 2014). These findings support our model, and they also offer an explanation for why deletion of EXO1 in a checkpoint-deficient background did not result in full recovery of damage bypass competence in our system.

### Controlling nuclease activity during replication

Our data suggest that uncontrolled Exo1 and to some extent Pif1 activity is largely responsible for the catastrophic loss of viability upon entry into S phase when both checkpoint signaling and damage bypass are inactivated. This raises the question of what structures need to be protected by the checkpoint in order to prevent chromosome fragmentation and lethality during replication of damaged templates. We cannot formally exclude collapsed forks as the origin of lethality, as Rad18 may normally operate both at daughter-strand gaps and "on-the-fly", that is, directly at the replication fork. However, we favor a scenario where checkpoint signaling is required primarily to maintain the integrity of daughter-strand gaps by preventing excessive resection (Fig 7B), because the synthetic lethality between checkpoint and damage bypass defects applies only to DNA damage, but not to HU-mediated fork problems and is observable only with rad9Δ, but not with mrc1Δ mutants. Moreover, daughter-strand gaps are the structures known to hyper-accumulate in the absence of damage bypass. Hence, it appears likely that the erosion of such structures is responsible for the loss of viability in our assay. A possible mechanism by which daughter-strand gaps could give rise to DSBs would be the merger of an expanding gap with a nick on the parental strand, possibly in the context of a nucleotide or base excision repair intermediate. Both damage bypass (via gap filling) and checkpoint signaling (via inhibition of gap expansion) are expected to counteract such events, which would explain why we observed DSBs only under conditions where both pathways are inactive. Alternatively, extended regions of ssDNA may simply be more vulnerable to attack by endonucleases or other endogenous sources of damage.

A dependence of viability on Rad9-mediated checkpoint signaling in damage bypass-deficient cells has also been noted in response to chronic low-dose exposure to genotoxic agents (Hishida et al, 2009; Huang et al, 2013). This phenomenon was attributed to an accumulation of daughter-strand gaps and even checkpoint-proficient cells gradually lost viability over time after such treatment. Deletion of EXO1 mitigated the effect to some extent, again arguing that control over Exo1 activity is particularly important to prevent erosion of daughter-strand gaps.

Intriguingly, another nuclease—Mre11—has been implicated in the degradation of nascent DNA in vertebrate systems (Hashimoto

et al, 2010; Schlacher et al, 2011). Here, protection of regressed forks or gap structures from Mre11-mediated resection was found to depend on homologous recombination factors such as BRCA1 and Rad51. While fork regression has not been observed in checkpoint-competent yeast cells, a spontaneous accumulation of daughter-strand gaps was also found in budding yeast rad52Δ and rad51Δ mutants (Hashimoto et al, 2010). It is possible that the aggravation of the exo1Δ checkpoint defect that we observed upon deletion of MRE11 is attributable to the same phenomenon.

In addition to Rad53-mediated inhibition of its activity, we found Exo1—like Pif1—to be strongly cell cycle-regulated. Thus, the nuclease apparently needs to be carefully controlled in order to balance its beneficial versus potentially detrimental actions. This may reflect the necessity to prevent homologous recombination in G1, but another intriguing possibility is that Exo1 regulation could contribute to controlling the balance between damage processing via template switching versus translesion synthesis: its downregulation at the end of S phase might inhibit template switching by preventing gap expansion, thus further promoting the temporal separation of the two pathways that is already suggested by the staggered expression patterns of Rad5 and Rev1 (Waters & Walker, 2006; Ortiz-Bazan et al, 2014).

Our observations demonstrate that the interplay between Rad53 and Exo1, ensuring robust checkpoint activation while at the same time protecting regions of ssDNA from harmful expansion, constitutes an important and very general aspect of the cellular damage response, promoting genome maintenance by such diverse pathways as homologous recombination at DSBs, long-patch NER and postreplicative damage bypass.

## Materials and Methods

### Yeast strains and growth conditions

All yeast strains are listed in Appendix Table S1. The doxycycline-repressible Tet-RAD18 construct has been described previously (Daigaku et al, 2010). Strains carrying gene deletions or epitope-tagged alleles were created by PCR-based methods or by mating and tetrad dissection. Degron-tagged alleles of RAD53 and POL1 were constructed as described (Morawska & Ulrich, 2013). All strains carrying a deletion or degron-tagged allele of RAD53, deletion of MEC1 or deletion of both RAD9 and MRC1 were constructed in an sml1Δ background. Strains carrying exo1-SA and exo1-SA-ND alleles were created as described previously (Doerfler & Schmidt, 2014). To test for dominance, the exo1-SA allele containing its own promoter region was introduced into the Tet-RAD18 strain on an integrative plasmid inserted at the URA3 locus. All strains were grown at 30°C in YPD or synthetic complete (SC) medium supplemented with the appropriate amino acids. Cells were synchronized in G1 using 10 μg/ml α-factor for 2 h. Auxin was used at 1 mM, doxycycline at 2 μg/ml.

### Analysis of DNA damage sensitivities and recovery of viability

Sensitivity to HU, MMS, and 4NQO was determined by spotting 10-fold serial dilutions of exponentially growing cultures onto YPD plates containing the indicated concentrations of damaging agents.

Plates were incubated at 30°C for 3 days before imaging. Recovery upon Rad18 induction was measured as described previously (Daigaku *et al*, 2010). Briefly, *Tet-RAD18* cells were pre-grown in YPD containing doxycycline, that is, under *RAD18*-repressing conditions. They were then synchronized in G1, UV-irradiated (20 J/m$^2$) and released in the presence of doxycycline to maintain *RAD18* repression. At the indicated times, aliquots at appropriate dilutions were plated onto medium with or without doxycycline to either maintain repression (−Rad18) or induce expression of *RAD18* (+Rad18), respectively. After 2–3 days, colonies were counted and survival was determined relative to unirradiated cultures plated without doxycycline. Standard deviations (SD) were calculated from at least three independent experiments with three technical replicates each. For most recovery experiments involving degron-tagged alleles, protein degradation was induced by adding auxin (Sigma) to the culture at the indicated times. Dilutions were then plated at the indicated times onto auxin-containing medium (+Aux) to maintain protein degradation with (−Rad18) or without doxycycline (+Rad18). For the experiment shown in Fig 3C, Rad53 degradation was induced by adding auxin to the culture at the time of α-factor addition. At the indicated times, dilutions were plated onto medium with or without auxin in the absence of doxycycline to induce *RAD18* expression (+Rad18).

**Pulsed-field gel electrophoresis**

As described above, G1-synchronized cultures of the indicated strains were UV-irradiated (20 J/m$^2$) and released into S phase in the presence (−Rad18) or absence (+Rad18) of doxycycline, and samples were collected at the indicated times after release. Total DNA from $2 \times 10^7$ cells was then extracted in low-melting agarose plugs as described previously (Bianco *et al*, 2012), and chromosomes were resolved by PFGE in 1% agarose gels at 8°C using CHEF DR III (Bio-Rad) under the following conditions: 175 V with 80 s pulse time for 12 h and 110 s pulse time for 12 h. Chromosomes V and IV were detected by Southern blots using specific probes.

**Detection of proteins**

Total yeast protein extracts were prepared by trichloroacetic acid (TCA) precipitation as described (Morawska & Ulrich, 2013). Following SDS/PAGE and Western blotting, Rad53 phosphorylation was detected using monoclonal anti-Rad53 antibody (kindly provided by S. Gasser). Monoclonal anti-tubulin antibody YL1/2 (Sigma) was used to detect Rnr4 as described (Tsaponina *et al*, 2011). 9Myc- and 6HA-tagged proteins were detected using monoclonal antibodies 9E10 (Enzo) and F7 (Santa Cruz), respectively. Monoclonal anti-Pgk1 antibody 22C5D8 (Invitrogen) was used for loading controls. For visualization of Exo1$^{9myc}$, Exo1$^{6HA}$ and Pif1$^{6HA}$ phosphorylation, proteins were resolved in Phos-tag gels [8% polyacrylamide (29:1), containing 50 μM Phos-tag reagent] prepared as described previously (Kinoshita *et al*, 2006).

**Fluorescence microscopy**

In order to visualize Rad52$^{YFP}$, cells carrying plasmid pWJ1213 were fixed with 2.5% formaldehyde in potassium phosphate at pH 6.4 for 10 min, washed twice with potassium phosphate at pH 6.6, and stored in potassium phosphate at pH 7.4. Cells were permeabilized with 80% ethanol for 10 min and resuspended in 0.5 μg/ml 4′,6-diamidino-2-phenylindole (DAPI). At least 250 cells derived from three independent experiments were analyzed for each time point. Images were acquired with a 63× objective on a wide-field fluorescence microscope (AF7000, Leica) equipped with an ORCA-Flash 4.0 V2 digital CMOS camera (Hamamatsu) under the control of LAS AF software (Leica). Images were processed with ImageJ software (https://imagej.nih.gov/ij/). To analyze spindle formation, immunofluorescence of tubulin was performed as described (Matos *et al*, 2013), and images were acquired with a 63× objective on an Axio Imager (ZEISS) equipped with a Hamamatsu CCD Camera under the control of Volocity software.

**DNA fiber analysis**

BrdU×7 cells (a kind gift from Philippe Pasero) were harvested in S phase after relevant treatments. Genomic DNA was prepared in agarose plugs as described previously (Bianco *et al*, 2012; Kaykov *et al*, 2016). Following proteinase K digestion and extensive washing in 1× TE with 100 mM NaCl, genomic DNA was extracted from plugs by melting in 50 mM MES, pH 6.0, with 100 mM NaCl. Combing was performed on vinylsilane-coated coverslips with a molecular combing system (Genomic Vision). ssDNA was detected by immunostaining with mouse anti-ssDNA antibody (MAB3034, Millipore) and Cy3-labeled goat anti-mouse secondary antibody (AB6946, Abcam). EdU tracts were labeled with Alexa Fluor 647 via Click reaction (Click-iT Plus Imaging Kit, Invitrogen). DNA fibers were counterstained with YOYO-1 and imaged with a Deltavision Elite System (GE Healthcare) equipped with a FITC/TRITC/Cy5 filter set. For analysis, binary images were obtained by applying absolute thresholds that were adjusted so as to eliminate non-specific signals within unreplicated (YOYO-1-positive), undamaged DNA. Identical parameters were chosen for each set of samples processed together, but parameters were adjusted individually for different sets of experiments. The lengths of individual EdU tracts as well as the number of ssDNA tracts and the total length of ssDNA within each of these regions were determined using the ImageJ FIJI software. The same analysis was performed on EdU-negative regions in the same set of fibers. The total percentage of ssDNA and the number of ssDNA tracts per unit length of DNA was then calculated and plotted for each tract. Statistical analysis was performed with GraphPad Prism v7.0.

**Expanded View** for this article is available online.

## Acknowledgements

We thank IMB's Microscopy core facility for advice with fluorescence microscopy, IMB's media laboratory for supplies, and John Diffley, Susan Gasser, Nicolas Hoch, Philippe Pasero, Félix Prado, Rodney Rothstein, Kristina Schmidt and Ralf Wellinger for providing reagents. This work was funded by Cancer Research UK and an Advanced Grant of the European Research Council (ERC AdG 323179: DAMAGE BYPASS).

## Author contributions

NG-R, MM, YD, and HDU conceived the study; NG-R, MM, RPW, and YD performed and analyzed experiments; NG-R and HDU wrote the manuscript; and all authors discussed the data and commented on the manuscript.

## Conflict of interest

The authors declare that they have no conflict of interest.

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
