## [Review Process File · The EMBO Journal]

Spatial separation between replisome- and template-induced replication stress signaling

Néstor García-Rodríguez, Magdalena Morawska, Ronald P. Wong, Yasukazu Daigaku and Helle D. Ulrich

Review timeline:

Submission date:	5 October 2017
Editorial Decision:	8 November 2017
Appeal received:	10 November 2017
Editorial Decision:	12 December 2017
Revision received:	29 January 2018
Accepted:	26 February 2018

Editor: Hartmut Vodermaier

Transaction Report:

1st Editorial Decision

8 November 2017

Thank you for submitting your manuscript on checkpoint signaling at daughter-strand gaps for our editorial consideration. We have now received the comments of three expert referees, who subsequently also further discussed the study amongst themselves. I am afraid to say that in light of these assessments and considerations, we had to conclude that the study is presently not a strong candidate for an EMBO Journal article. While the referees do not question the quality of the present data, especially reviewers 1 and 3 feel that many of the described observations have already been noted previously in various contexts; and although they appreciate the potential interest of the newly derived repriming model, they remain unconvinced that the current data set provides sufficiently definitive support for this new model and in disfavor of alternative scenarios. Given that these issues were also acknowledged by the (generally nevertheless positive) second reviewer during our post-review referee discussions, we are unfortunately not convinced that the study could easily be revised towards becoming suitable for a broad general journal such as The EMBO Journal, and therefore decided not to invite a resubmission on this occasion.

Thank you in any case for having had the opportunity to consider this work. I am sorry that the outcome of this evaluation was not a more positive one, but hope that you will still find our referees' detailed comments and suggestions helpful for proceeding further with this study. In any case, thank you again for the opportunity to consider this work.

REFEREE REPORTS

Referee #1:

Garcia-Rodríguez et al. dissect how daughter-strand gaps formed by replication-blocking lesions in PRR-defective mutants activate checkpoint signalling and contribute to cell survival in budding yeast. They demonstrate that Rad9, and not Mrc1, is the primary checkpoint-adaptor that activates Rad53 in S phase during replication stress induced by DNA lesions. Failure to activate the

checkpoint in RAD18 deficient cells following UV treatment results in fragmented chromosomes, Rad52 foci, an irregular G2/M and ultimately cell death. Through the use of inducible expression/degradation alleles they show that for cells to survive Rad53 must be activated during the S phase where damage is encountered while Rad18 activation can be delayed until G2. They tested many downstream targets of Rad53 searching for the mediator of survival and identified phosphorylation of Exo1, and to a lesser extent Pif1, as key targets. They show that Exo1 and Pif1 are phosphorylated in S phase following UV treatment and that Exo1 is required for S phase Rad53 activation in response to replication-blocking lesions but not replisome-inherent problems. Finally, using DNA combing, they reveal Exo1-dependent ssDNA gaps associated with replicated DNA during lesion-induced replication stress.

The authors propose a model where the nature of how a replication forks stalls dictates the checkpoint adapters involved and subsequent downstream cellular response. During replisome-inherent stresses such as limiting dNTPs or genetic perturbations a Mrc1-dependant checkpoint is utilized, while during replisome blocking lesions such as the ones resulting from MMS or UV treatment a Rad9- and Exo1-dependent checkpoint is preferred.

The main contributions of this work include: additional evidence that Exo1 is a key Rad53 target in S phase that mediates survival of PRR defective cells during lesion-induced fork stalling, a negative feedback loop where Exo1 resection activates Rad53 which in turn downregulates Exo1 and inhibits toxic levels of resection, and a role for Rad9 in lesion-induced checkpoint activation during S-phase. While there is clear genetic evidence for their conclusions the study lacks mechanistic detail, particularly how Rad9 and Mrc1 are differentially regulated during different fork-stalling scenarios. Additionally, a number of key concepts here have been noted before. The role of Exo1 as toxic in the absence of Rad53 in MMS is clear in Segurado and Diffley-I think the novelty here is limited to the Rad53 activation side of the feedback loop (although the concept is the same as introduced in Morin et al. 2008 for uncapped telomeres), and the similarity to resection events at DSBs and NER gaps, and the role of MRX in checkpoint activation, make it not entirely surprising that Exo1 is important for Rad53 activation in MMS. Regulation of Exo1 by Rad53 is well-known (Morin et al). Segurado and Diffley also cover the notion that uncontrolled Exo1 is largely responsible for viability loss when checkpoint signalling is compromised at the level of Rad53. The authors could be clearer about what the inactivation of bypass adds (is there anything other than making the sensitivity and other phenotypes more penetrant?). The activation of Rad53 via Rad9 in response to template lesions (MMS) has been noted by Smolka (Ohouo et al., Cussiol et al) limiting the novelty of the Mrc1 vs Rad9 argument. That lesion-induced stress is detected behind the fork is noted in Balint et al. The measurement of Exo1-dependent gaps, and the timing experiments with Rad53 activation are both quite interesting. I would like to see more integration with the existing MMS/fork stress literature and a clearer delineation of how the current work unifies and extends some of these existing concepts.

Specific comments:

- Fig 1A: Doses of 2 and 5 mM HU used in the spot assay result in extreme sensitivity of the rad53Δsml1Δ and mec1sml1Δ strains in +Rad18. This leaves no dynamic range to test for increased sensitivity in the -Rad18 scenario. The authors should address this by using a lower dose of HU that permits growth rad53Δsml1Δ and mec1sml1Δ strains. Conversely, rad18Δ alone is sensitive to higher doses of HU in spot assays and there appears to be a slight increase of sensitivity in rad18Δmrc1Δ and rad18Δrad9Δ at 5 mM HU (FigS1A). The authors should use higher doses of HU to test for increased sensitivity of rad9Δ and mrc1Δ +/- Rad18.
- Fig 4: The exo1Δ and pif1Δ rescues of the rad53Δ phenotypes, while significant, are still an order of magnitude below the RAD53 scenario. While addressed briefly in the discussion this should also be acknowledged in the results section.
- Fig 6: The authors should compare % ssDNA in WT, rad18Δ and rad18Δ exo1Δ cells during UV and MMS treatment in order to provide some evidence that Exo1 expands gaps resulting from absence of bypass.
- Fig 6: It is unclear how the % ssDNA is calculated. It appears that % ssDNA equals the total length of ssDNA tracks divided by the total length of EdU tracks. The mode of measurement should be clearly stated in the text, figure legend and materials and methods section.
- Fig 6: The amount of DNA replication in the HU treatment will be significantly lower than in the Control, MMS and UV treated samples. To adequately compare the formation of ssDNA the authors should use a lower dose of HU that allow for comparable lengths of replication tracks.

Referee #2:

In this manuscript, Garcia-Rodriguez and colleagues address the very important question of how the Mec1-Rad53 checkpoint pathway is activated in response to replication stress. Textbook models show that single-stranded DNA (ssDNA) is generated by uncoupling of leading strand and lagging strand synthesis when replication forks encounter DNA lesions induced by MMS or UV light. However, this model is largely based on assumptions and direct evidence showing that ssDNA is produced at forks is still lacking. Here, the authors show that daughter strand gaps can be created behind the fork by repriming after the lesions and expansion of the resulting ssDNA gaps by the Exo1 exonuclease. They also provide evidence that these gaps activate the Rad9-dependent branch of the intra-S phase checkpoint, which is distinct from the Mrc1-dependent branch involved in the detection of fork stalling induced by replication inhibitors such as hydroxyurea or aphidicolin. Overall, this manuscript is well written and the data are of very high quality. In particular, the DNA fibers showing that ssDNA gaps accumulate within replicated tracks in an Exo1-dependent manner after MMS and UV exposure, but not HU, are very convincing. These data shed new light on the mechanisms by which cells detect replication impediments induced by different types of genotoxic drugs and provide a mechanistic ground for the interplay between the Rad9 and Mrc1 branches of the Mec1-Rad53 pathway. Since mechanisms are conserved from yeast to human and are highly relevant to cancer development and drug resistance mechanisms, this study should be of wide general interest. However, some issues need to be addressed prior to publication.

Specific issues:

1. Page 4, last paragraph: The statement that template switching can be uncoupled from replication fork progression "without adverse effects" may be too strong. Indeed studies from the Sale lab indicate that this uncoupling can interfere with the transmission of epigenetic information and alter gene expression profiles.
2. Page 9, first paragraph and Fig. 2A: The fact that rad53 rad18 double mutants (but not rad53 mutants) undergo massive chromosome fragmentation is intriguing. The authors could speculate on the mechanism involved.
3. The Western blot showing that Pif1 is phosphorylated by Rad53 (Fig. 4B) is not convincing. The authors should use a Phos-tag gel, as for Exo1 as in Fig. S6E.

Referee #3:

This manuscript shows that replication of UV and MMS-damaged DNA generate ssDNA behind the replication forks, which is processed by Exo1 to generate gaps that are sensed by the checkpoint protein Rad9. The authors propose that replication of UV and MMS damaged DNA generates a checkpoint response triggered by post-replicative daughter strand gaps, whereas the checkpoint induced by HU treatment originates from stalled replication fork.

While the experiments are well performed, some of the results are not particularly novel. The finding that Exo1 is inhibited by Rad53 has been previously shown by D. Lydall. That the response to HU is different from that to MMS and UV has already established and this manuscript does not provide convincing results to explain this difference. The interpretation of the results by the authors is, from my point of view, not very convincing and alternative interpretations could be given (fork stalling/collapse, Exo1-mediated processing, DSBs and Rad9 activation). Experiments aimed at verifying fork collapse, repriming and Exo1-mediated enlargement of daughter strand gaps are essential to support the authors' model.

Specific points:

1. In figure 1A, the authors cannot conclude that Rad18 depletion does not sensitize mec1, rad53 and mrc1 rad9 mutants to HU as mec1, rad53 and mrc1 rad9 mutants are already dead at 2mM HU even in the presence of functional Rad18.
2. Since UV-induced lesions generated in G1 are known to activate a Rad9-dependent checkpoint that delays the G1/S transition, the authors need to show in figure 1 that a checkpoint response is

induced in these conditions (Rad53 phosphorylation) and that this checkpoint is different from the canonical Rad9-dependent G1/S checkpoint (NER dependency).

3. Since cells lacking Rad18 are slow growing, other translesion synthesis should be tested to exclude that the effect is specific to Rad18 depletion.
4. In Figure 5, Rad53 phosphorylation after MMS treatment is induced immediately after G1 release indicating that a G1/S rather than an S phase checkpoint is activated. Thus, the possible Exo1-mediated processing of DNA gaps, leading to RAD9-dependent checkpoint activation, is similar to what has been previously reported in non-replicating cells. The authors need to use a lower MMS and demonstrate that cells are in S phase and not in G1 when the checkpoint is activated (by budding profile and NOT by FACS).
5. In figure 6, the authors need to analyze cell cycle progression and checkpoint activation. How the authors conclude that the ssDNA tracts they detect are daughter-strand gaps is not clear to me. They could be generated simply by Exo1-mediated processing of newly synthesized strands rather than from Exo1-mediated enlargement of gaps generated by repriming events. This point is crucial to support the model.
6. The authors claim (see also the title) that "checkpoint signaling originates from Exo1-processing of daughter-strand gaps". The finding that the lack of Rad18 does not increase checkpoint activation does not fit with the model. It should do, since the gaps are not filled-in and Exo1 can process them. An alternative model is that MMS or UV treatment leads to fork collapse and exposure to Exo1-mediated processing. This could lead to the generation of DSBs that can activate Rad9.
7. The finding that EXO1 deletion suppresses the MMS sensitivity of rad53 but not of mec1 mutants (Segurado and Diffley, 2005) or that RAD9 deletion increases the HU sensitivity of rad53 and mec1 mutants does not fit with the model.

Author Appeal

10 November 2017

Thank you for conveying the reviewers' reports to us. As you can imagine, we were very disappointed to learn that we would not be allowed a revision of our manuscript. I understand that the reviewers' opinions about the novelty and quality of our results are mixed, but I think that this is due to some serious misconceptions, in particular on the side of reviewer #3. I would therefore appreciate if you would consider the following arguments for a moment:

- All reviewers comment on the high quality of the data, suggesting that they do not have a problem with the results per se.
- Reviewer #1 and #2 accept our model, whereas reviewer #3 is not convinced. He/she essentially suggests that a) cells may not be in S phase during checkpoint activation or b) cells experience double-strand breaks during replication over damaged templates, and this is what activates the checkpoint. We have very clearly ruled out scenario a) (and additionally quote relevant literature to support us), and postulating that sub-lethal doses of UV radiation cause double-strand breaks seems so far-fetched that we consider this highly unlikely - indeed, our data provide evidence to the contrary. In addition, this reviewer does not appear to have read our manuscript carefully, as he/she suggests several experiments that we do show. Thus, his/her criticisms are completely unjustified.
- What remains is the problem that reviewers #1 and #3 admonish a lack of novelty of our study because some of the individual observations (Rad53 activation via Rad9 versus Mrc1 and Exo1 activities at NER gaps or DSBs) are known from other cellular contexts. This even leads reviewer #1 to claim that our findings are "not surprising". This, I believe, is at the core of the negative criticisms; however, both reviewers have overlooked that the message of our study is not simply a re-capitulation of the Exo1-Rad53 relationship in another context - this would indeed not be very novel - but instead a new explanation for the dichotomy of checkpoint signalling by replisome versus template damage. The notion that checkpoint activation via Mec1 and Rad53 requires two distinct mediators, Mrc1 and Rad9, depending on the type of replication stress, has been known for a long time, but why this is the case has never been explained. Our data for the first time provide a

satisfying model that resolves this conundrum. Textbook models of replication stress invariably invoke stalled replication forks as the origin of the checkpoint signal - but we now show that this does not apply to common types of damaging agents such as UV radiation. The fact that our model is very consistent internally and with the published literature makes it easy to label it - with hindsight! - as "not surprising", but this does not make it in any way less novel or worth publishing, as it has not been formulated before.

As you see, I am strongly convinced that our study actually provides fundamentally new insight into the mechanism of the replication stress response that should be important and interesting for a broad readership and should change the textbook models. I have presented these results repeatedly at meetings and seminars and have every time received comments such as "this is an entirely new view on the damage checkpoint" - by experts in the field. I would therefore urge you to reconsider your decision to reject this manuscript outright. I have appended a response to the reviewers' comments in which I have explained our reasoning in more detail, and if possible, I would appreciate if you would take the time for a further discussion of our viewpoints on the phone.

I look forward to hearing from you.

Referee #1:

Garcia-Rodriguez et al. dissect how daughter-strand gaps formed by replication-blocking lesions in RR-defective mutants activate checkpoint signalling and contribute to cell survival in budding yeast. They demonstrate that Rad9, and not Mrc1, is the primary checkpoint-adapter that activates Rad53 in S phase during replication stress induced by DNA lesions. Failure to activate the checkpoint in RAD18 deficient cells following UV treatment results in fragmented chromosomes, Rad52 foci, an irregular G2/M and ultimately cell death. Through the use of inducible expression/degradation alleles they show that for cells to survive Rad53 must be activated during the S phase where damage is encountered while Rad18 activation can be delayed until G2. They tested many downstream targets of Rad53 searching for the mediator of survival and identified phosphorylation of Exo1, and to a lesser extent Pif1, as key targets. They show that Exo1 and Pif1 are phosphorylated in S phase following UV treatment and that Exo1 is required for S phase Rad53 activation in response to replication-blocking lesions but not replisome-inherent problems. Finally, using DNA combing, they reveal Exo1-dependent ssDNA gaps associated with replicated DNA during lesion-induced replication stress.

Unfortunately, this summary omits a rather important point: the observation that checkpoint activation by damaged templates in wildtype cells follows the same mechanism as observed in bypass-defective mutants. This significantly broadens the scope of our study.

The authors propose a model where the nature of how a replication forks stalls dictates the checkpoint adapters involved and subsequent downstream cellular response. During replisome-inherent stresses such as limiting dNTPs or genetic perturbations a Mrc1-dependant checkpoint is utilized, while during replisome blocking lesions such as the ones resulting from MMS or UV treatment a Rad9- and Exo1-dependent checkpoint is preferred.

Our most important message is not reflected here: the notion that replisome(Mrc1)- and template(Rad9)-induced problems are sensed and processed in a spatially distinct manner, i.e. at or behind replication forks, respectively.

The main contributions of this work include: additional evidence that Exo1 is a key Rad53 target in S phase that mediates survival of PRR defective cells during lesion-induced fork stalling, a negative feedback loop where Exo1 resection activates Rad53 which in turn downregulates Exo1 and inhibits toxic levels of resection, and a role for Rad9 in lesion-induced checkpoint activation during S-phase. While there is clear genetic evidence for their conclusions the study lacks mechanistic detail, particularly how Rad9 and Mrc1 are differentially regulated during different fork-stalling scenarios.

We strongly disagree with the statement that we do not provide mechanistic insight. On the contrary, we have obtained detailed mechanistic insight into how the ssDNA that activates the

damage checkpoint during DNA replication is generated: by Exo1 activity on daughter-strand gaps! We have even visualised this activity directly by means of DNA combing. The differential uses of Rad9 and Mrc1 then follow from their spatial arrangement: Mrc1 as a replisome component senses replisome problems, while Rad9, by means of its binding to methylated histone H3, acts within chromatinised regions.

Additionally, a number of key concepts here have been noted before. The role of Exo1 as toxic in the absence of Rad53 in MMS is clear in Segurado and Diffley-I think the novelty here is limited to the Rad53 activation side of the feedback loop (although the concept is the same as introduced in Morin et al. 2008 for uncapped telomeres), and the similarity to resection events at DSBs and NER gaps, and the role of MRX in checkpoint activation, make it not entirely surprising that Exo1 is important for Rad53 activation in MMS. Regulation of Exo1 by Rad53 is well-known (Morin et al). Segurado and Diffley also cover the notion that uncontrolled Exo1 is largely responsible for viability loss when checkpoint signalling is compromised at the level of Rad53.

The reviewer is right that many of these observations have been noted in other contexts before, and we have duly cited appropriate references. The purpose of this manuscript is not to simply recapitulate these experiments in yet another system, but in order to characterise checkpoint activation in our situation, it was unavoidable to verify these observations in our context. Importantly, the conclusion that we derive from those data about the Exo1-mediated activation of damage signalling upon encounter of damaged templates as opposed to replisome problems is novel and completely unrelated to the contexts in which the Rad53-Exo1 feedback loop has previously been described. Once all the pieces have been put together, it may in hindsight not be surprising that Exo1 works as it does in other situations; but this connection and the implications have in fact not been formulated by anyone before.

The authors could be clearer about what the inactivation of bypass adds (is there anything other than making the sensitivity and other phenotypes more penetrant?).

As we explain in the text, inactivation of bypass has indeed served the purpose of hyper-accumulating daughter-strand gaps, thus making the dependence of damage signalling on these structures more penetrant. Importantly, however, we have then moved on to show that the mechanism of Exo1-dependent Rad53 activation also applies to the wildtype situation.

The activation of Rad53 via Rad9 in response to template lesions (MMS) has been noted by Smolka (Ohouo et al., Cussiol et al) limiting the novelty of the Mrc1 vs Rad9 argument.

The distinction between Mrc1- vs. Rad9-mediated signalling and their relationship to replisome-versus template-induced damage has been postulated by others, but importantly, we are the first to provide a satisfying mechanistic explanation for this dichotomy. This is the core of the message of our study.

That lesion-induced stress is detected behind the fork is noted in Balint et al.

Balint et al. have exclusively focussed on MMS-induced damage, and they do not at all consider the origin of the ssDNA that induces damage signalling behind the fork. They describe their results as generally applicable to checkpoint activation during replication, thus completely neglecting the conflicting results of fork-associated checkpoint activation by conditions such as HU treatment. Hence, their data do not allow to draw any conclusion about the spatial separation between replisome- and template-induced checkpoint signalling as we do.

The measurement of Exo1-dependent gaps, and the timing experiments with Rad53 activation are both quite interesting. I would like to see more integration with the existing MMS/fork stress literature and a clearer delineation of how the current work unifies and extends some of these existing concepts.

These experiments, shown in Figures 5 and 6, form the gist of our study, and we realise that it might be somewhat taxing to put the reader through all that comes before. However, in order to fully characterise the mechanism of checkpoint activation at gaps, we needed the Tet-Rad18 system as a model case where those gaps accumulate before being able to assess the situation in wildtype cells.

By putting forth our model of spatially segregated checkpoint activation by replisome- versus template-associated replication stress signalling and linking this to the previous notion of Mrc1- versus Rad9-mediated checkpoint activation, we have succeeded in integrating a lot of previously disjointed literature about replication stress and unifying this into a coherent model for the first time. In addition, we have reconciled the notion of re-priming with the concept of fork uncoupling. We find it hard to see how these data could be even more integrated with prior concepts, but we are of course open to suggestions.

Specific comments:

- Fig 1A: Doses of 2 and 5 mM HU used in the spot assay result in extreme sensitivity of the rad53Δsm11Δ and mec1sm11Δ strains in +Rad18. This leaves no dynamic range to test for increased sensitivity in the -Rad18 scenario. The authors should address this by using a lower dose of HU that permits growth rad53Δsm11Δ and mec1sm11Δ strains. Conversely, rad18Δ alone is sensitive to higher doses of HU in spot assays and there appears to be a slight increase of sensitivity in rad18Δmrc1Δ and rad18Δrad9Δ at 5 mM HU (FigS1A). The authors should use higher doses of HU to test for increased sensitivity of rad9Δ and mrc1Δ +/- Rad18.

This is doable.

- Fig 4: The exo1Δ and pif1Δ rescues of the rad53Δ phenotypes, while significant, are still an order of magnitude below the RAD53 scenario. While addressed briefly in the discussion this should also be acknowledged in the results section.

This is doable.

- Fig 6: The authors should compare % ssDNA in WT, rad18Δ and rad18Δ exo1Δ cells during UV and MMS treatment in order to provide some evidence that Exo1 expands gaps resulting from absence of bypass.

This is doable in principle, although we consider it much more important to show this effect in wildtype cells, as the rad18-deficient cells only served as a model case for gap accumulation.

- Fig 6: It is unclear how the % ssDNA is calculated. It appears that % ssDNA equals the total length of ssDNA tracks divided by the total length of EdU tracks. The mode of measurement should be clearly stated in the text, figure legend and materials and methods section.

This is explained in the methods section, but we can add it to the text and figure legend.

- Fig 6: The amount of DNA replication in the HU treatment will be significantly lower than in the Control, MMS and UV treated samples. To adequately compare the formation of ssDNA the authors should use a lower dose of HU that allow for comparable lengths of replication tracks.

This is doable.

Referee #2:

In this manuscript, Garcia-Rodriguez and colleagues address the very important question of how the Mec1-Rad53 checkpoint pathway is activated in response to replication stress. Textbook models show that single-stranded DNA (ssDNA) is generated by uncoupling of leading strand and lagging strand synthesis when replication forks encounter DNA lesions induced by MMS or UV light. However, this model is largely based on assumptions and direct evidence showing that ssDNA is produced at forks is still lacking. Here, the authors show that daughter strand gaps can be created behind the fork by repriming after the lesions and expansion of the resulting ssDNA gaps by the Exo1 exonuclease. They also provide evidence that these gaps activate the Rad9-dependent branch of the intra-S phase checkpoint, which is distinct from the Mrc1-dependent branch involved in the detection of fork stalling induced by replication inhibitors such as hydroxyurea or aphidicolin. Overall, this manuscript is well written and the data are of very high quality. In particular, the DNA fibers showing that ssDNA gaps accumulate within replicated tracks in an Exo1-dependent manner

after MMS and UV exposure, but not HU, are very convincing. These data shed new light on the mechanisms by which cells detect replication impediments induced by different types of genotoxic drugs and provide a mechanistic ground for the interplay between the Rad9 and Mrc1 branches of the Mec1-Rad53 pathway. Since mechanisms are conserved from yeast to human and are highly relevant to cancer development and drug resistance mechanisms, this study should be of wide general interest. However, some issues need to be addressed prior to publication.

We appreciate this reviewer's comments on the importance of our findings. The reviewer correctly points out that the notion of fork uncoupling is generally assumed to be applicable to all types of replication stress without much evidence, and that our study illustrates the importance to differentiate between different types of challenges.

Specific issues:

1. Page 4, last paragraph: The statement that template switching can be uncoupled from replication fork progression "without adverse effects" may be too strong. Indeed studies from the Sale lab indicate that this uncoupling can interfere with the transmission of epigenetic information and alter gene expression profiles.

This is doable.

2. Page 9, first paragraph and Fig. 2A: The fact that rad53 rad18 double mutants (but not rad53 mutants) undergo massive chromosome fragmentation is intriguing. The authors could speculate on the mechanism involved.

This is doable.

3. The Western blot showing that Pif1 is phosphorylated by Rad53 (Fig. 4B) is not convincing. The authors should use a Phos-tag gel, as for Exo1 as in Fig. S6E.

This is doable.

Referee #3:

This manuscript shows that replication of UV and MMS-damaged DNA generate ssDNA behind the replication forks, which is processed by Exo1 to generate gaps that are sensed by the checkpoint protein Rad9. The authors propose that replication of UV and MMS damaged DNA generates a checkpoint response triggered by post-replicative daughter strand gaps, whereas the checkpoint induced by HU treatment originates from stalled replication fork.

While the experiments are well performed, some of the results are not particularly novel. The finding that Exo1 is inhibited by Rad53 has been previously shown by D. Lydall. That the response to HU is different from that to MMS and UV has already established and this manuscript does not provide convincing results to explain this difference.

As explained above in our response to reviewer #1, we do not claim that we were the first to describe these relationships, but we strongly disagree with the notion that our data do not provide convincing results to explain this difference, as outlined below.

The interpretation of the results by the authors is, from my point of view, not very convincing and alternative interpretations could be given (fork stalling/collapse, Exo1-mediated processing, DSBs and Rad9 activation).

We do not think it is fair to simply assert that we have not excluded alternative explanations without providing details of what these should entail. In our discussion, we have very carefully considered all the alternative checkpoint activation models that have been discussed in the literature and that we considered reasonable and relevant to our conditions. The reviewer appears to suggest an alternative model where checkpoint activation by UV or MMS damage routinely requires fork

stalling, fork collapse and DSBs as obligatory intermediates. We consider this highly unlikely for the following reasons:

- *fork stalling: we cite data from the Costanzo lab and others that have readily observed daughter-strand gaps, but very little strand uncoupling in replicating MMS-treated DNA, and we as well as many other labs (including Jentsch, Lopez, Marian, Kelly) have convincingly demonstrated that re-priming is a common phenomenon. This concept, which postulates that forks do not necessarily stall upon encountering a lesion is now very well established.*
- *fork collapse: it is very well established that under our conditions (e.g. a single dose 20 J/m² UV radiation), where survival of repair- and checkpoint-competent cells is close to 100%, forks do not collapse. In fact, fork collapse upon UV treatment is not at all commonly known in checkpoint-proficient cells, and much more drastic conditions (e.g. treatment with a combination of high doses of HU and MMS) or checkpoint inactivation are needed to cause a break-down of the replisome. Hence, postulating repeated fork collapse and re-assembly in order to generate the postreplicative gaps that we observe appears to us a much more involved and unnecessarily complicated model than to invoke the well-established concept of re-priming.*
- *DSBs: As we show in our Figure 2, we observe widespread DSBs only when damage bypass and checkpoint signalling are both deficient. It is true that we cannot exclude a minute fraction of wildtype cells experiencing DSBs; however, this would not trigger an overall checkpoint activation in all cells. Sub-lethal doses of UV in asynchronous cultures (which include S phase cells!) have certainly never been reported to result in DSBs in wildtype cells. Again, this appears to us a much more convoluted model than postulating a gap-associated, Exo1-dependent mechanism analogous to what is observed at NER gaps.*

Experiments aimed at verifying fork collapse, repriming and Exo1-mediated enlargement of daughter strand gaps are essential to support the authors' model.

We have directly verified Exo1-mediated enlargement of daughter strand gaps by DNA fibre analysis (Fig. 6). Our previous work has demonstrated re-priming in the same experimental system that we use here (Daigaku et al., Nature 2010), and we have quoted this work here. Our fibre analysis also demonstrates that the ssDNA regions are situated within tracts of newly synthesized DNA, which is inconsistent with irreversible fork collapse. I am not sure how the reviewer expects us to distinguish re-priming from repeated reversible fork collapse and re-start. However, we consider generation of ssDNA gaps via re-priming a more parsimonious explanation for our data.

Specific points:

1. In figure 1A, the authors cannot conclude that Rad18 depletion does not sensitize mec1, rad53 and mrc1 rad9 mutants to HU as mec1, rad53 and mrc1 rad9 mutants are already dead at 2mM HU even in the presence of functional Rad18.

There is actually residual growth at 2 mM HU in the strains mentioned here, but it is not a problem to repeat the experiment at lower HU concentrations.

2. Since UV-induced lesions generated in G1 are known to activate a Rad9-dependent checkpoint that delays the G1/S transition, the authors need to show in figure 1 that a checkpoint response is induced in these conditions (Rad53 phosphorylation) and that this checkpoint is different from the canonical Rad9-dependent G1/S checkpoint (NER dependency).

We show exactly this in Figure 5. We show that at the doses used here, release into S phase is needed to activate the checkpoint (Fig. 5G), and we quoted published results demonstrating that NER-deficient cells can activate the checkpoint in S phase like wildtype cells.

3. Since cells lacking Rad18 are slow growing, other translesion synthesis should be tested to exclude that the effect is specific to Rad18 depletion.

We do not feel that this is necessary, as Rad18 depletion only serves as a means to hyper-accumulate gaps, and we verify the same effects in wildtype cells later on in the study.

4. In Figure 5, Rad53 phosphorylation after MMS treatment is induced immediately after G1 release indicating that a G1/S rather than an S phase checkpoint is activated. Thus, the possible Exo1-mediated processing of DNA gaps, leading to RAD9-dependent checkpoint activation, is similar to what has been previously reported in non-replicating cells. The authors need to use a lower MMS and demonstrate that cells are in S phase and not in G1 when the checkpoint is activated (by budding profile and NOT by FACS).

The Diffley lab has shown very convincingly that MMS does not activate the checkpoint in G1 cells, but in contrast that replication is needed for this to happen (Tercero et al., Mol. Cell 2003). We have considered this point and have quoted the paper in the discussion. Moreover, Figure 5G clearly shows that the checkpoint is not activated when cells are not released after exposure to damage.

5. In figure 6, the authors need to analyze cell cycle progression and checkpoint activation. How the authors conclude that the ssDNA tracts they detect are daughter-strand gaps is not clear to me. They could be generated simply by Exo1-mediated processing of newly synthesized strands rather than from Exo1-mediated enlargement of gaps generated by repriming events. This point is crucial to support the model.

If the ssDNA tracts were derived from fork-associated Exo1-processing, they should always be found at the ends of tracts of newly replicated DNA, but never internally. Internal gaps therefore indicate re-priming. The only alternative would be the merging of neighbouring replicons with a lesion in the middle. However, considering that we performed these experiments in early S phase where replication tracts are short these events should be rare.

6. The authors claim (see also the title) that "checkpoint signaling originates from Exo1-processing of daughter-strand gaps". The finding that the lack of Rad18 does not increase checkpoint activation does not fit with the model. It should do, since the gaps are not filled-in and Exo1 can process them.

This reviewer appears to misinterpret our data here to imply that lack of Rad18 does not increase checkpoint activation. In fact, Figure 5 shows the opposite: it demonstrates rather clearly that in the absence of Rad18, all of Rad53 becomes phosphorylated within 20 min (Fig. 5A), whereas in the presence of Rad18, Rad53 phosphorylation is only partial (Fig. 5C). We showed a similar effect earlier (Daigaku et al., Nature 2010). Thus, these observations fully support our model.

An alternative model is that MMS or UV treatment leads to fork collapse and exposure to Exo1-mediated processing. This could lead to the generation of DSBs that can activate Rad9.

As already explained above, the notion that MMS or UV treatment at sub-lethal doses causes fork collapse and DSBs in repair- and checkpoint-proficient cells is unprecedented and much more complicated than invoking re-priming and gap-expansion, and we have no evidence to suggest that DSBs are involved.

7. The finding that EXO1 deletion suppresses the MMS sensitivity of rad53 but not of mec1 mutants (Segurado and Diffley, 2005) or that RAD9 deletion increases the HU sensitivity of rad53 and mec1 mutants does not fit with the model.

Mec1 has additional targets that are independent of Rad53. Hence, it is not surprising that EXO1 deletion does not suppress mec1 mutants. An additive effect between rad9 and rad53 or mec1 is not the subject of our study, and we do not show any experiments pertaining to this relationship.

Thank you very much for your patience during our reconsideration of your manuscript, and apologies for the delay in getting back to you. I was indeed waiting for feedback from an arbitrating advisor, whom I had given access to your paper, the reviewers' comments as well as your responses to them. Our advisor has now come back with their assessment, which I am pleased to say was generally favorable (see excerpt from his/her comments copied below). We shall therefore be happy to revert our original decision and consider this work further for eventual publication in The EMBO Journal, pending satisfactory revision to address the originally raised issues via textual clarification

and/or additional experiments. In general, these revisions should go along the lines you already proposed in your point-by-point rebuttal letter, but in addition, I feel it would be helpful to also include some further experiments to directly answer referee 3's concerns about the MMS doses currently used causing G1 checkpoint-activating damage, as this point came up also during the referees' original pre-decision consultation (ref 3 points 4 and 5 - maybe adding data with lower MMS doses and/or checking cell cycle progression and checkpoint activation?).

As usual, it is our policy to consider only a single round of major revision (making it important to satisfactorily respond to all key points in this round), during which any competing manuscripts published elsewhere will have no negative impact on our final decision on your study. Please do not hesitate to contact me should you have any additional questions in this regard. I look forward to receiving your revision.

COMMENTS FROM ARBITRATING ADVISOR:

In my opinion, the paper is very solid and convincing. The major finding that - template-induced post-replicative daughter strand gaps are created behind the fork by repriming events and that there is expansion of the resulting ssDNA by Exo1 in a Rad9-dependent manner, is novel - and they clearly show that this is distinct from the Mrc1 (replisome-induced) checkpoint signaling. This is novel as well. The DNA combing is convincing, and they provide extensive mechanistic insight into the phenomena.

Reviewers 1 & 3 did not seem to get the point of the paper. The authors should stress that in their conditions DSBs are not detectable in wt, rad53, or rad18 mutants when the checkpoint is fully activated; DSBs are detectable only in the rad18 rad53 double. This helps argue against a DSB-driven checkpoint, and is supported by other data in their paper.

EXCERPT FROM REFEREE 2's ORIGINAL CROSS-COMMENTS:

"... I also agree with referee #3 that the authors cannot formally exclude the possibility that the gaps shown in Fig.6 are due to resection of nascent DNA and not to repriming. This possibility could be experimentally addressed, for instance in the absence of Mre11. ..."

EXCERPT FROM REFEREE 3's ORIGINAL CROSS-COMMENTS (AFTER REF 2):

"...The gaps they detect in figure 6 are not necessarily due to repriming and processing and can be explained by processing without repriming. For this reason, I don't think that the lack of Mre11 (or of other nucleases) can discriminate between the two possibilities. (...) My other concern is that the MMS dose they use is very high and is known to activate a G1 checkpoint (see Rad53 that is immediately phosphorylated after release). In this case, the gaps are not generated by repriming events in S phase but just by processing around MMS or UV-induced lesions and this has been already published by M. Muzi Falconi in response to UV irradiation in G1 and G2 cells. "

Response to the Reviewers' Comments

We thank the reviewers for their constructive comments, in particular the arbitrating advisor for their strong support regarding the novelty and soundness of our study. By performing a series of additional experiments, including an expanded DNA fibre analysis, and revising critical sections of the text, we have now addressed the reviewers' concerns. In order to convey our message more clearly, we have also revised the title of our manuscript. For added clarity, all revisions have been highlighted in colour in the revised manuscript. A detailed response to the reviewers' comments is given below.

Referee #1:

Garcia-Rodriguez et al. dissect how daughter-strand gaps formed by replication-blocking lesions in RR-defective mutants activate checkpoint signalling and contribute to cell survival in budding yeast. They demonstrate that Rad9, and not Mrc1, is the primary checkpoint-adaptor that activates Rad53 in S phase during replication stress induced by DNA lesions. Failure to activate the checkpoint in RAD18 deficient cells following UV treatment results in fragmented chromosomes, Rad52 foci, an irregular G2/M and ultimately cell death. Through the use of inducible expression/degradation alleles they show that for cells to survive Rad53 must be activated during the S phase where damage is encountered while Rad18 activation can be delayed until G2. They tested many downstream targets of Rad53 searching for the mediator of survival and identified phosphorylation of Exo1, and to a lesser extent Pif1, as key targets. They show that Exo1 and Pif1 are phosphorylated in S phase following UV treatment and that Exo1 is required for S phase Rad53 activation in response to replication-blocking lesions but not replisome-inherent problems. Finally, using DNA combing, they reveal Exo1-dependent ssDNA gaps associated with replicated DNA during lesion-induced replication stress.

*Unfortunately, this summary omits a rather important point: the observation that checkpoint activation by damaged templates **in wildtype cells** follows the same Exo1-dependent mechanism as in bypass-defective mutants. This significantly broadens the scope of our study.*

The authors propose a model where the nature of how a replication forks stalls dictates the checkpoint adapters involved and subsequent downstream cellular response. During replisome-inherent stresses such as limiting dNTPs or genetic perturbations a Mrc1-dependant checkpoint is utilized, while during replisome blocking lesions such as the ones resulting from MMS or UV treatment a Rad9- and Exo1-dependent checkpoint is preferred.

*Our most important message is not reflected here: the notion that replisome (Mrc1)- and template (Rad9)-induced problems are sensed and processed **in a spatially distinct manner**, i.e. at or behind replication forks, respectively. In order to convey this message more clearly, we have changed the title of our manuscript.*

The main contributions of this work include: additional evidence that Exo1 is a key Rad53 target in S phase that mediates survival of PRR defective cells during lesion-induced fork stalling, a negative feedback loop where Exo1 resection activates Rad53 which in turn downregulates Exo1 and inhibits toxic levels of resection, and a role for Rad9 in lesion-induced checkpoint activation during S-phase. While there is clear genetic evidence for their conclusions the study lacks mechanistic detail, particularly how Rad9 and Mrc1 are differentially regulated during different fork-stalling scenarios.

*We strongly disagree with the statement that we do not provide mechanistic insight. On the contrary, we have obtained **detailed mechanistic insight** into how the ssDNA that activates the*

damage checkpoint during DNA replication is generated: by Exo1 activity on daughter-strand gaps! We have even visualised this activity directly by means of DNA combing. The differential uses of Rad9 and Mrc1 then follow from their spatial arrangement: Mrc1 as a replisome component senses replisome problems, while Rad9, by means of its binding to methylated histone H3, acts within chromatinised regions.

Additionally, a number of key concepts here have been noted before. The role of Exo1 as toxic in the absence of Rad53 in MMS is clear in Segurado and Diffley-I think the novelty here is limited to the Rad53 activation side of the feedback loop (although the concept is the same as introduced in Morin et al. 2008 for uncapped telomeres), and the similarity to resection events at DSBs and NER gaps, and the role of MRX in checkpoint activation, make it not entirely surprising that Exo1 is important for Rad53 activation in MMS. Regulation of Exo1 by Rad53 is well-known (Morin et al). Segurado and Diffley also cover the notion that uncontrolled Exo1 is largely responsible for viability loss when checkpoint signalling is compromised at the level of Rad53.

*The reviewer is right that many of these observations have been noted **in other contexts** before, and we have duly cited appropriate references. The purpose of this manuscript is not to simply recapitulate these experiments in yet another system, but in order to characterise checkpoint activation in our situation, it was unavoidable to verify these observations in our context. Importantly, the conclusion that we derive from those data about the Exo1-mediated activation of damage signalling **upon encounter of damaged templates as opposed to replisome problems** is novel and unrelated to the contexts in which the Rad53-Exo1 feedback loop has previously been described. Once all the pieces have been put together, it may in hindsight not be surprising that Exo1 works as it does in other situations; but this connection and the implications have in fact not been formulated before.*

The authors could be clearer about what the inactivation of bypass adds (is there anything other than making the sensitivity and other phenotypes more penetrant?).

*As we explain in the text, inactivation of bypass has indeed served the purpose of hyper-accumulating daughter-strand gaps, thus making the dependence of damage signalling on these structures more penetrant. Importantly, however, we have then moved on to show that the **mechanism of Exo1-dependent Rad53 activation also applies to the wildtype situation.***

The activation of Rad53 via Rad9 in response to template lesions (MMS) has been noted by Smolka (Ohouo et al., Cussiol et al) limiting the novelty of the Mrc1 vs Rad9 argument.

*The distinction between Mrc1- vs. Rad9-mediated signalling and their relationship to replisome stalling (by HU) versus DNA damage (by MMS) has first been described in detail by Alcasabas et al. (Nat Cell Biol 2001). Many other studies, such as those mentioned by the reviewer, have successfully built on those results by describing in more detail the modulation of Rad9 activity. Importantly, however, we are the **first to provide a satisfying mechanistic explanation for this dichotomy.** This is the core of the message of our study, and we have tried to emphasize this more clearly in the revised title, the introduction and the discussion sections.*

That lesion-induced stress is detected behind the fork is noted in Balint et al.

Balint et al. indeed show in their contribution how the Slx4-Rtt107 complex, an important checkpoint regulator that contributes to full activation of Mec1, assembles at “discrete

domains distal to the stressed replication forks". This is a significant study that supports our results, and we now refer to it in the introduction and discussion. However, their study focusses entirely on MMS-induced damage and is therefore unable to resolve the conundrum of parallel Mrc1- and Rad9-mediated signalling pathways and the conflicting results obtained with HU treatment. They also do not elaborate on the nature of the domains where the Slx4-Rtt107 complex assembles, i.e. they do not clarify whether these are indeed ssDNA. Hence, their data do not allow to draw any conclusion about the spatial separation between replisome- and template-induced checkpoint signalling as we do here.

The measurement of Exo1-dependent gaps, and the timing experiments with Rad53 activation are both quite interesting. I would like to see more integration with the existing MMS/fork stress literature and a clearer delineation of how the current work unifies and extends some of these existing concepts.

These experiments, shown in Figures 5 and 6, form the gist of our study, and we realise that it might be somewhat taxing to put the reader through all that comes before. However, in order to fully characterise the mechanism of checkpoint activation at gaps, we needed the Tet-Rad18 system as a model case where those gaps accumulate before being able to assess the situation in wildtype cells.

*By putting forth our **model of spatially segregated checkpoint activation** by replisome- versus template-associated replication stress signalling and **linking this to the previous notion of Mrc1- versus Rad9-mediated checkpoint activation**, we have succeeded in integrating a lot of previously disjointed literature about replication stress and unifying this into a coherent model for the first time. Importantly, we have **reconciled the two contradictory concepts of re-priming and fork uncoupling**. We find it hard to see how these data could be even more integrated with prior concepts, but we are of course open to suggestions.*

Specific comments:

- Fig 1A: Doses of 2 and 5 mM HU used in the spot assay result in extreme sensitivity of the rad53Δsml1Δ and mec1sml1Δ strains in +Rad18. This leaves no dynamic range to test for increased sensitivity in the -Rad18 scenario. The authors should address this by using a lower dose of HU that permits growth rad53Δsml1Δ and mec1sml1Δ strains. Conversely, rad18Δ alone is sensitive to higher doses of HU in spot assays and there appears to be a slight increase of sensitivity in rad18Δmrc1Δ and rad18Δrad9Δ at 5 mM HU (FigS1A). The authors should use higher doses of HU to test for increased sensitivity of rad9Δ and mrc1Δ +/- Rad18.

We show here for the reviewers the results of an extended titration of HU in both genetic backgrounds. Our data indicate that the removal of Rad18 adds very little sensitivity to that of the checkpoint mutants by themselves – certainly not enough to call synergism. Accordingly, we have replaced the images of the HU sensitivity tests in Fig. 1A and S1A (now EV1A) with new panels from the figure below, showing sensitivities at 1 and 2 mM HU.

The reviewer is also concerned about the sensitivity of the rad18Δ mutant at high concentrations of HU. Panel B (below) indeed shows a moderate sensitivity of rad18Δ towards high concentrations of HU, which is slightly enhanced by deletion of any of the checkpoint genes. However, this enhancement is again rather mild compared to what is observed with MMS or 4NQO, and importantly there is no difference between the effects of mrc1Δ and rad9Δ. In order to visualise this properly, we have added new panels from the figure below, showing a higher HU dose, to Figure S1A (now EV1A).

*In addition, we have now modified the text to point out that "depletion of Rad18 **only mildly enhanced** the sensitivity of any of the strains towards **moderate concentrations of HU**".*

- Fig 4: The *exo1Δ* and *pif1Δ* rescues of the *rad53Δ* phenotypes, while significant, are still an order of magnitude below the *RAD53* scenario. While addressed briefly in the discussion this should also be acknowledged in the results section.

Done.

- Fig 6: The authors should compare % ssDNA in WT, *rad18Δ* and *rad18Δ* *exo1Δ* cells during UV and MMS treatment in order to provide some evidence that Exo1 expands gaps resulting from absence of bypass.

As explained above, the rad18-deficient cells only served as a model case for gap accumulation, and we would like to make our point more broadly, arguing that Exo1 expands daughter-strand gaps in general, not only under conditions of hyperaccumulation. We have therefore performed all experiments with Rad18-positive cells. These data further support the notion that daughter-strand gaps routinely arise behind forks even in a WT setting.

- Fig 6: It is unclear how the % ssDNA is calculated. It appears that % ssDNA equals the total length of ssDNA tracks divided by the total length of EdU tracks. The mode of measurement should be clearly stated in the text, figure legend and materials and methods section.

This was explained in the methods section, but we have now expanded the description in the figure legend and the main text.

- Fig 6: The amount of DNA replication in the HU treatment will be significantly lower than in the Control, MMS and UV treated samples. To adequately compare the formation of ssDNA the authors should use a lower dose of HU that allow for comparable lengths of replication tracks.

We have repeated the experiment with half the dose of HU (60 mM for 60 min). Under these conditions, replication tracts are comparable in length to those of control cells that were released for 30 min in the absence of damage. Our data show very similar results, i.e. no increase (but rather a decrease) in the percentage of ssDNA in HU. We have added these data as a supplementary Figure (Appendix Fig. S3A-C).

Referee #2:

In this manuscript, Garcia-Rodriguez and colleagues address the very important question of how the Mec1-Rad53 checkpoint pathway is activated in response to replication stress. Textbook models show that single-stranded DNA (ssDNA) is generated by uncoupling of leading strand and lagging strand synthesis when replication forks encounter DNA lesions induced by MMS or UV light. However, this model is largely based on assumptions and direct evidence showing that ssDNA is produced at forks is still lacking. Here, the authors show that daughter strand gaps can be created behind the fork by repriming after the lesions and expansion of the resulting ssDNA gaps by the Exo1 exonuclease. They also provide evidence that these gaps activate the Rad9-dependent branch of the intra-S phase checkpoint, which is distinct from the Mrc1-dependent branch involved in the detection of fork stalling induced by replication inhibitors such as hydroxyurea or aphidicolin. Overall, this manuscript is well written and the data are of very high quality. In particular, the DNA fibers showing that ssDNA gaps accumulate within replicated tracks in an Exo1-dependent manner after MMS and UV exposure, but not HU, are very convincing. These data shed new light on the mechanisms by which cells detect replication impediments induced by different types of genotoxic drugs and provide a mechanistic ground for the interplay between the Rad9 and Mrc1 branches of the Mec1-Rad53 pathway. Since mechanisms are conserved from yeast to human and are highly relevant to cancer development and drug resistance mechanisms, this study should be of wide general interest. However, some issues need to be addressed prior to publication.

*We appreciate this reviewer's comments on the importance of our findings. The reviewer correctly points out that the notion of fork uncoupling is generally **assumed to be applicable to all types of replication stress without much evidence**, and that our study illustrates the importance to **differentiate between different types of challenges**.*

Specific issues:

1. Page 4, last paragraph: The statement that template switching can be uncoupled from replication fork progression "without adverse effects" may be too strong. Indeed studies from the Sale lab indicate that this uncoupling can interfere with the transmission of epigenetic information and alter gene expression profiles.

*We have modified the statement to a more cautious one, quoting the data from the Sales lab: "They can be delayed without **major effects on genome stability** until bulk genome replication is completed (Daigaku et al, 2010; Karras & Jentsch, 2010; Ulrich, 2009), **although an impact on the transmission of epigenetic information has been reported (Sarkies et al, 2010).**"*

2. Page 9, first paragraph and Fig. 2A: The fact that rad53 rad18 double mutants (but not rad53 mutants) undergo massive chromosome fragmentation is intriguing. The authors could speculate on the mechanism involved.

We have included a brief discussion of this issue in the discussion section:

“A possible mechanism by which daughter-strand gaps could give rise to DSBs would be the merger of an expanding gap with a nick on the parental strand, possibly in the context of a nucleotide or base excision repair intermediate. Both damage bypass (via gap filling) and checkpoint signalling (via inhibition of gap expansion) would counteract such events, which would explain why we observed DSBs only under conditions where both pathways are inactive. Alternatively, extended regions of ssDNA may simply be more vulnerable to attack by endonucleases or other endogenous sources of damage.”

3. The Western blot showing that Pif1 is phosphorylated by Rad53 (Fig. 4B) is not convincing. The authors should use a Phos-tag gel, as for Exo1 as in Fig. S6E.

Fig. 4B was actually generated from a Phos-tag gel. We have repeated the experiment and replaced the figure by what we believe is a better image, although a number of higher-MW bands appearing above Pif1 at later time points (possibly representing additional phosphorylation sites) still interfere with a “clean” picture of Pif1. Nevertheless, the non-phosphorylated forms of Pif1 remain prevalent in the rad53 mutant, whereas virtually all of Pif1 is shifted to higher MW forms in the WT, thus again supporting our conclusion. Moreover, Rossi et al. (Cell Rep 2015) have published very similar data about Rad53-dependent Pif1 phosphorylation in response to replication stress induced by HU.

Referee #3:

This manuscript shows that replication of UV and MMS-damaged DNA generate ssDNA behind the replication forks, which is processed by Exo1 to generate gaps that are sensed by the checkpoint protein Rad9. The authors propose that replication of UV and MMS damaged DNA generates a checkpoint response triggered by post-replicative daughter strand gaps, whereas the checkpoint induced by HU treatment originates from stalled replication fork.

While the experiments are well performed, some of the results are not particularly novel. The finding that Exo1 is inhibited by Rad53 has been previously shown by D. Lydall. That the response to HU is different from that to MMS and UV has already established and this manuscript does not provide convincing results to explain this difference.

As explained above in our response to reviewer #1, we do not claim that we were the first to describe these relationships, but we strongly disagree with the notion that our data do not provide convincing results to explain this difference, as outlined below.

The interpretation of the results by the authors is, from my point of view, not very convincing and alternative interpretations could be given (fork stalling/collapse, Exo1-mediated processing, DSBs and Rad9 activation).

The reviewer appears to suggest an alternative model where checkpoint activation by UV or MMS damage routinely requires fork stalling, fork collapse and DSBs as obligatory intermediates. We consider this highly unlikely for the following reasons:

- fork stalling: we cite data from the Costanzo lab and others that have readily observed daughter-strand gaps, but very little strand uncoupling in replicating MMS-treated DNA, and many labs (including Jentsch, Lopez, Marian, Kelly and our own) have by now convincingly demonstrated that re-priming is a common phenomenon. This concept, which postulates that **forks do not necessarily stall upon encountering a lesion** is now very well established.
- fork collapse: it is widely accepted that under our conditions (e.g. a single dose 20 J/m² UV radiation), where survival of **repair- and checkpoint-competent cells** is close to 100%, **forks do not collapse**. In fact, fork collapse upon UV treatment is not at all commonly known in checkpoint-proficient cells, and much more drastic conditions (e.g. treatment with a combination of high doses of HU and MMS) or checkpoint inactivation are needed to cause a break-down of the replisome. Hence, postulating repeated fork collapse and re-assembly in order to generate the postreplicative gaps that we observe appears to us a much more involved and unnecessarily complicated model than to invoke the well-established concept of re-priming.
- DSBs: As we show in our Figure 2, we observe DSBs only when damage bypass and checkpoint signalling are both deficient. It is true that we cannot exclude a small fraction of wildtype cells experiencing DSBs; however, this would not trigger an overall checkpoint activation in all cells. Sub-lethal doses of UV in asynchronous cultures (which include S phase cells!) have certainly not been reported to result in DSBs in wildtype cells. Again, this appears to us a much more convoluted model than postulating a gap-associated, Exo1-dependent mechanism analogous to what is observed at NER gaps.

Experiments aimed at verifying fork collapse, repriming and Exo1-mediated enlargement of daughter strand gaps are essential to support the authors' model.

Our **previous work has demonstrated re-priming** in the same experimental system that we use here (Daigaku et al., Nature 2010), and we have quoted this work here. We have **directly verified Exo1-mediated enlargement of daughter strand gaps** by DNA fibre analysis (Fig. 6). This analysis also demonstrates that the ssDNA regions are situated within tracts of newly synthesized DNA, which is **inconsistent with irreversible fork collapse**. I am not sure how the reviewer expects us to distinguish re-priming from repeated reversible fork collapse and re-start. However, we consider generation of ssDNA gaps via re-priming a more parsimonious explanation for our data, not least because re-priming has by now been well established by several labs.

Specific points:

1. In figure 1A, the authors cannot conclude that Rad18 depletion does not sensitize mec1, rad53 and mrc1 rad9 mutants to HU as mec1, rad53 and mrc1 rad9 mutants are already dead at 2mM HU even in the presence of functional Rad18.

As also suggested by reviewer #1, we have repeated the experiment at lower HU concentrations and have replaced the corresponding panels in Fig. 1A. Please see also the image showing a more extensive titration of HU concentrations in our response to reviewer #1 above.

2. Since UV-induced lesions generated in G1 are known to activate a Rad9-dependent checkpoint that delays the G1/S transition, the authors need to show in figure 1 that a checkpoint response is induced in these conditions (Rad53 phosphorylation) and that this checkpoint is different from the canonical Rad9-dependent G1/S checkpoint (NER dependency).

We address exactly this point in Figure 5. We show that at the doses used here, release into S phase is needed to activate the checkpoint (Fig. 5G). Activation of the checkpoint in G1 in fact appears to require significantly higher doses; e.g. Muzi-Falconi and coworkers used 75 J/m² in their assays demonstrating Exo1 activity on NER gaps (Giannattasio et al, Mol Cell 2010). In support of our results, the Longhese lab reported “that Rad53 phosphorylation of rad14Δ cells UV-irradiated in G1 requires entry into S phase” (Neecke et al., EMBO J 1999). These data provide further support for an NER-independent mechanism of S phase-associated checkpoint activation. We refer to these observations in our discussion.

3. Since cells lacking Rad18 are slow growing, other translesion synthesis should be tested to exclude that the effect is specific to Rad18 depletion.

We do not think that this is necessary, as Rad18 depletion only serves as a means to hyper-accumulate gaps, and we verify the same effects in wildtype cells later on in the study, implying that the effect is not specific to rad18-depleted cells.

4. In Figure 5, Rad53 phosphorylation after MMS treatment is induced immediately after G1 release indicating that a G1/S rather than an S phase checkpoint is activated. Thus, the possible Exo1-mediated processing of DNA gaps, leading to RAD9-dependent checkpoint activation, is similar to what has been previously reported in non-replicating cells. The authors need to use a lower MMS and demonstrate that cells are in S phase and not in G1 when the checkpoint is activated (by budding profile and NOT by FACS).

The Diffley lab has shown very convincingly that MMS does not activate the checkpoint in G1 cells, but that replication is needed for this to happen (Tercero et al., Mol. Cell 2003). We have have quoted the paper in the discussion. For the reviewer’s information, we insert Figure 4C from Tercero et al. (2003) here:

(C) MMS treatment in G1 activates Rad53 in the subsequent S phase. W303-1a cells were blocked in G1 with α factor and held in G1 for an extra hour in the presence of 0.033% MMS. The culture was split in two; half was released into S phase in the absence of MMS and the other held in G1 with α factor for 1 hr without MMS. Samples were taken at the time points indicated.

The conditions used in that experiment correspond to an overall dosage very similar to what we used (0.08% for 30 min). Moreover, our new Figure 5H clearly shows that the checkpoint is not activated when cells are not released into S phase after exposure to MMS. Nevertheless, in response to the reviewer’s concern, we have repeated our assay at a lower MMS concentration (0.04%) and included budding ratios as well as a blot showing degradation of the cyclin inhibitor Sic1 as a cell cycle marker to verify entry into S phase. Although overall checkpoint activation is lower at this MMS dose, both the rise of the budding index and the degradation of Sic1 coincide well with the appearance of phosphorylated Rad53 in WT cells (30-40 min), whereas checkpoint activation is reduced and delayed until ca. 60 min in the exo1 mutant. We have added these data as Appendix Fig. S2.

5. In figure 6, the authors need to analyze cell cycle progression and checkpoint activation. How the authors conclude that the ssDNA tracts they detect are daughter-strand gaps is not clear to me. They could be generated simply by Exo1-mediated processing of newly synthesized strands rather than from Exo1-mediated enlargement of gaps generated by repriming events. This point is crucial to support the model.

If the ssDNA tracts were derived replication-independently from expanded NER gaps, a similar pattern should be visible outside the EdU tracts. We have therefore added a quantification of ssDNA tracts in the un-replicated regions (Fig. 6C, D). This shows significantly fewer and shorter gaps than inside the EdU tracts, indicating that the pattern of ssDNA within the newly replicated region is distinct from that of un-replicated DNA and thus follows specifically from replication.

This leaves the formal possibility that the ssDNA tracts arose from fork-associated Exo1-processing of nascent DNA. In this case, however, they should always be found at the ends of tracts of newly replicated DNA, but not internally. Internal gaps therefore indicate re-priming unless they resulted from the merging of neighbouring replicons with a lesion in the middle. However, considering that we performed these experiments in early S phase where replication tracts are short, these events should be rare.

We are therefore confident that the majority of ssDNA within the newly replicated regions indeed represents daughter-strand gaps. We have revised the text to explain this.

6. The authors claim (see also the title) that "checkpoint signaling originates from Exo1-processing of daughter-strand gaps". The finding that the lack of Rad18 does not increase checkpoint activation does not fit with the model. It should do, since the gaps are not filled-in and Exo1 can process them.

This reviewer appears to misinterpret our data here to imply that lack of Rad18 does not increase checkpoint activation. In fact, Figure 5 shows the opposite: it demonstrates rather clearly that in the absence of Rad18, all of Rad53 becomes phosphorylated within 20 min (Fig. 5A), whereas in the presence of Rad18, Rad53 phosphorylation is only partial (Fig. 5C). We showed a similar effect earlier (Daigaku et al, Nature 2010). Thus, these observations fully support our model.

An alternative model is that MMS or UV treatment leads to fork collapse and exposure to Exo1-mediated processing. This could lead to the generation of DSBs that can activate Rad9.

As already explained above, the notion that MMS or UV treatment at sub-lethal doses causes fork collapse and DSBs in repair- and checkpoint-proficient cells is unprecedented and much more complicated than invoking re-priming and gap expansion. Importantly, we have no evidence to suggest that DSBs are involved unless both damage bypass and checkpoint signalling are inactivated (Fig. 2A).

7. The finding that EXO1 deletion suppresses the MMS sensitivity of rad53 but not of mec1 mutants (Segurado and Diffley, 2005) or that RAD9 deletion increases the HU sensitivity of rad53 and mec1 mutants does not fit with the model.

Mec1 has additional targets that are independent of Rad53. Hence, it is not surprising that EXO1 deletion does not suppress mec1 mutants. An additive effect between rad9 and rad53 or mec1 is not the subject of our study, and we do not show any experiments pertaining to this relationship.

COMMENTS FROM ARBITRATING ADVISOR:

In my opinion, the paper is very solid and convincing. The major finding that - template-induced post-replicative daughter strand gaps are created behind the fork by repriming events and that there is expansion of the resulting ssDNA by Exo1 in a Rad9-dependent manner, is novel - and they clearly show that this is distinct from the Mrc1 (replisome-induced) checkpoint signaling . This is novel as well. The DNA combing is convincing, and they provide extensive mechanistic insight into the phenomena. Reviewers 1 & 3 did not seem to get the point of the paper. The authors should stress that in their conditions DSBs are not detectable in wt, rad53, or rad18 mutants when the checkpoint is fully activated; DSBs are detectable only in the rad18 rad53 double. This helps argue against a DSB-driven checkpoint, and is supported by other data in their paper.

We thank the advisor for their supportive and insightful comments. We have followed their advice and now further stress our observation that DSBs occur only when Rad53 and Rad18 are both absent, thus arguing for a DSB-independent mechanism of checkpoint activation.

EXCERPT FROM REFEREE 2's ORIGINAL CROSS-COMMENTS:

"... I also agree with referee #3 that the authors cannot formally exclude the possibility that the gaps shown in Fig.6 are due to resection of nascent DNA and not to repriming. This possibility could be experimentally addressed, for instance in the absence of Mre11. ..."

Please see our response to point 5 by reviewer #3.

EXCERPT FROM REFEREE 3's ORIGINAL CROSS-COMMENTS (AFTER REF 2):

"...The gaps they detect in figure 6 are not necessarily due to repriming and processing and can be explained by processing without repriming. For this reason, I don't think that the lack of Mre11 (or of other nucleases) can discriminate between the two possibilities. (...)

Please see our response to point 5 by reviewer #3.

My other concern is that the MMS dose they use is very high and is known to activate a G1 checkpoint (see Rad53 that is immediately phosphorylated after release). In this case, the gaps are not generated by repriming events in S phase but just by processing around MMS or UV-induced lesions and this has been already published by M. Muzi Falconi in response to UV irradiation in G1 and G2 cells. "

As explained above in our response to point 5 by reviewer #3, we have now expanded our DNA combing analysis in order to address the reviewer's concern that the ssDNA tracts might arise from replication-independent processing around lesions rather than in a postreplicative manner:

- 1. We have performed EdU-labelling followed by DNA combing in cells that have not been released from G1 after MMS treatment. Comparison with S phase cells shows that the vast majority of the EdU tracts represent genuine replication tracts (Appendix Fig. S3D, E).*
- 2. As explained in our response to point 5 of reviewer #3, we now show a quantification of ssDNA tracts not only in the EdU-labelled, newly replicated DNA, but also in the*

unreplicated regions of the S phase cells, i.e. in those regions of the fibres that were not EdU-labelled. Figures 6C and D shows that the increase in ssDNA is much less prominent in these regions than in the replicated DNA, and that the tracts are shorter. This result supports our notion that the majority of ssDNA within the newly replicated DNA is in fact replication-dependent, while the low percentage of ssDNA that is also observable in the unreplicated regions may in fact represent replication-independent NER events.

Thank you for submitting your revised manuscript for our consideration. It has now been seen once more by one of the original referees (see comments below), and I am happy to inform you that there are no further objections towards publication in The EMBO Journal.

Before we will be able to send you a formal letter of acceptance, there are just a few editorial issues that I would need you to address:

- Referee 2 asks for the discussion of one more recent reference (see below).
- Pre-acceptance checks by our data editors have raised several queries regarding data descriptors in the figure legends. We also require a brief Conflict of Interest Statement in the manuscript text.
- Please send us a modified Appendix PDF, in which the internal nomenclature has been adjusted from "Supplemental Table/Figure 1..." to "Appendix Table/Figure S1...", and in which the blue marking-up of modified text has been removed.
- Figure EV2 requires scale bars in micrographs.
- Finally, I would like to ask you to send us figure source data for the gels and blots in Figures 4B, 5, EV1B, EV5 and App. Figs. S1 and S2. We would ask for a single PDF/JPG/GIF file per figure comprising the original, uncropped and unprocessed scans of all such panels displayed in the respective figures. These should be labelled with the appropriate figure/panel number, and should have molecular weight markers; further annotation would clearly be useful but is not essential. They would be linked as such to the respective figures in the online publication of your article. Easiest is probably if you simply send them via one or several email messages.

Once we will have received the modified text, Appendix and EV2 figure files, as well as the raw data files, we should then hopefully be able to swiftly proceed with formal acceptance and production of the manuscript.

REFEREE REPORT

Referee #2 (Report for Author)

This revised version of the manuscript entitled « Spatial separation between replisome- and template-induced replication stress signaling » by Garcia-Rodriguez and colleagues presents compelling evidence that Rad53 is activated by Rad9 at template-induced postreplicative gaps, through a mechanism that is distinct from the Mrc1-dependent activation at stalled forks. I agree with the authors and the arbitrating advisor that this study provides novel mechanistic insights and integrates previously published (conflicting) results into a plausible model of checkpoint activation at damaged forks. In particular, this study argues against the view that fork uncoupling/hyper-unwinding represents the only way to generate ssDNA at arrested forks, as indicated in all textbook models. From this respect, this study represents a major advance in the field and should be of wide general interest. In my opinion, this manuscript is now suitable for publication in EMBO Journal.

Minor issue: The authors should cite a recent study from the Zhao lab (Hang et al., 2015, Mol. Cell 60, 268) showing that the repression of late origins in MMS-treated cells depends on Mrc1. This would support the view that the Mrc1 pathway is activated in the presence of MMS, even though it is dispensable for viability.

Corresponding Author Name: Ulrich

Journal Submitted to: The EMBO Journal

Manuscript Number: 98369